# The importance of species interactions in eco-evolutionary community dynamics under climate change

Anna Åkesson[1], Alva Curtsdotter [2], Anna Eklöf [1], Bo Ebenman[1], Jon Norberg [3] & György Barabás [1,4 ✉]

Eco-evolutionary dynamics are essential in shaping the biological response of communities to ongoing climate change. Here we develop a spatially explicit eco-evolutionary framework which features more detailed species interactions, integrating evolution and dispersal. We include species interactions within and between trophic levels, and additionally, we incorporate the feature that species' interspecific competition might change due to increasing temperatures and affect the impact of climate change on ecological communities. Our modeling framework captures previously reported ecological responses to climate change, and also reveals two key results. First, interactions between trophic levels as well as temperature-dependent competition within a trophic level mitigate the negative impact of climate change on biodiversity, emphasizing the importance of understanding biotic interactions in shaping climate change impact. Second, our trait-based perspective reveals a strong positive relationship between the within-community variation in preferred temperatures and the capacity to respond to climate change. Temperature-dependent competition consistently results both in higher trait variation and more responsive communities to altered climatic conditions. Our study demonstrates the importance of species interactions in an eco-evolutionary setting, further expanding our knowledge of the interplay between ecological and evolutionary processes.

[1] Division of Theoretical Biology, Dept. IFM, Linköping University, Linköping, Sweden. [2] Insect Ecology Lab, Zoology, The University of New England, Armidale, NSW, Australia. [3] Department of Ecology, Environment and Plant Sciences Stockholm University, Stockholm, Sweden. [4] MTA-ELTE Theoretical Biology and Evolutionary Ecology Research Group, Budapest, Hungary. ✉email: gyorgy.barabas@liu.se

Changing climatic conditions influence species' ecology, such as demography, biotic interactions, and movement, as well as species' evolutionary rates. Despite the acknowledgment of the highly important interplay between ecological and evolutionary processes for determining species distributions and survival under altered climatic conditions[1–4], few studies account for their combined effects[5]. The interplay between these processes has been partly addressed in previous work, showing unexpected results: inclusion of evolution potentially results in increased extinction rates when combined with dispersal[6], and high dispersal rates do not reduce extinctions since colonization often comes at expense of other species[7]. Moreover, species interactions can both alleviate and aggravate the impact of climate change on species[8], and interact with other eco-evolutionary processes. For example, species interactions can affect a species' evolutionary response to altered environmental conditions[9,10]; and dispersal may release a species from negative interactions through migration[11] or increase them through invasion[12].

These very promising eco-evolutionary studies, striving to include a variety of relevant biological mechanisms, commonly depict species interactions in a simplified manner. For example, Norberg et al.[7], Lasky[13], and Thompson and Fronhofer[6] take all of the aforementioned aspects into consideration, and their models are important stepping-stones along the way to map out the relevance of species interactions under dispersal, evolution, and climate change. However, they only consider competitive interactions, and even those under more[7] or less[6,13] restrictive assumptions: Norberg et al.[7], for instance, set all intra- and interspecific interaction strengths to be equal, and Lasky[13] uses diffuse competition, whereby species share one common intra- and another common interspecific competition coefficient.

To further develop the field of eco-evolutionary studies of species' response to climatic change, here we present a spatially explicit eco-evolutionary framework centered around a more detailed view of species interactions, interplaying with species' abilities to adapt to and disperse across local environments. We focus on two extensions. First, we include interactions both within and between trophic levels. Second, we consider the feature that species' interspecific competition might change due to increasing temperatures, and affect the impact of climate change on ecological communities.

Such temperature-dependent competition between species has usually not been considered in an eco-evolutionary setting[14–16]. Temperature-dependent competition is centered around the idea that each patch in the larger spatial landscape consists of multiple microhabitats, each with a somewhat higher or lower local temperature than the patch average. If the temperature optima of two competing species are similar, they will compete for the same microhabitats and thus experience strong competition. Similarly, competition will decrease if two species within a patch have different temperature optima. A temperature optimum mismatching the local mean temperature will result in a decreased local growth rate, but might still be favorable if it results in decreased interspecific competition.

Dispersal between and competitive interactions within local environments over time transfer to regional and global changes of ecological communities, and it is repeatedly shown that the effects of climate change differ between geographical regions[17]. As we here consider the globe's full latitudinal range from the polar regions to the equator, we can use our framework to evaluate how effects of local interactions and climate change vary depending on the region considered. We explore the effect of refined species interactions on (1) local trends (within each patch), including local species diversity; (2) regional trends (division of patches into polar, temperate, and tropical areas), including species' ranges and turnover; and (3) global trends, including global losses and the general community-wide capacity to respond to climatic change.

We show that influence from a second trophic level and, in particular, temperature-dependent competition affect both species distributions and global trends, giving higher levels of coexistence, lower levels of species turnover, and fewer global extinctions. Also, the interplay between ecological (e.g., dispersal and species interactions) and evolutionary (e.g., adaptation to new conditions) processes along a spatial gradient do significantly affect species' responses to altered climatic conditions in unexpected ways. For example, when species are able to both disperse and evolve fast, temperature-dependent competition results in more global losses than when the capacity to disperse and evolve is low. Furthermore, we demonstrate that community-wide dispersion of species' temperature optima is a strong predictor of a community's capacity to respond to climate change, which has implications for future management guidelines.

## Results

**Modeling framework.** We consider $S$ species distributed in $L$ distinct habitat patches. The patches form a linear latitudinal chain going around the globe, with dispersal between adjacent patches (Fig. 1). The state variables are species' local densities and local temperature optima (the temperature at which species achieve maximum intrinsic population growth). This temperature optimum is a trait whose evolution is governed by quantitative genetics[18–22]: each species, in every patch, has a normally distributed temperature optimum with a given mean and variance. The variance is the sum of a genetic and an environmental contribution. The genetic component is given via the infinitesimal model[23,24], whereby a very large number of loci each contribute a small additive effect to the trait. This has two consequences. First,

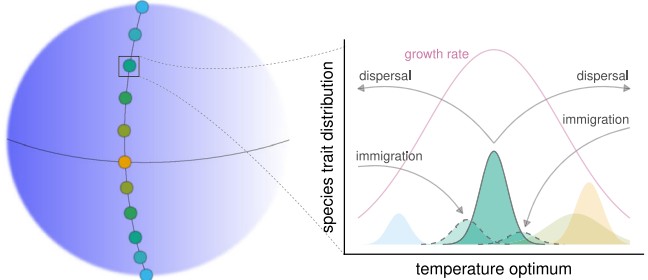

**Fig. 1 Illustration of our modeling framework.** There are several patches hosting local communities, arranged linearly along a latitudinal gradient. Patch color represents the local average temperature, with warmer colors corresponding to higher temperatures. The graph depicts the community of a single patch, with four species present. They are represented by the colored areas showing the distributions of their temperature optima, with the area under each curve equal to the population density of the corresponding species. The green species is highlighted for purposes of illustration. Each species has migrants to adjacent patches (independent of local adaptedness), as well as immigrants from them (arrows from and to the green species; the distributions with dashed lines show the trait distributions of the green species' immigrant individuals). The purple line is the intrinsic growth rate of a phenotype in the patch, as a function of its local temperature optimum (this optimum differs across patches, which is why the immigrants are slightly maladapted to the temperature of the focal patch.) Both local population densities and local adaptedness are changed by the constant interplay of temperature-dependent intrinsic growth, competition with other species in the same patch, immigration to or emigration from neighboring patches, and (in certain realizations of the model) pressure from consumer species.

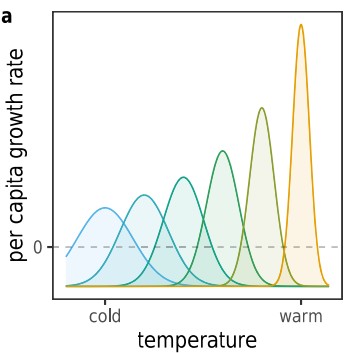
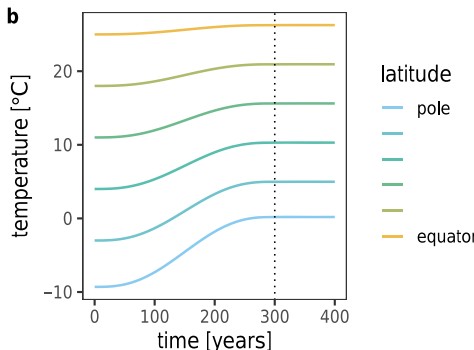

**Fig. 2 Temperature optima and climate curves. a** Different growth rates at various temperatures. Colors show species with different mean temperature optima, with warmer colors corresponding to more warm-adapted species. The curves show the maximum growth rate achieved when a phenotype matches the local temperature, and how the growth rate decreases with an increased mismatch between a phenotype and local temperature, for each species. The dashed line shows zero growth: below this point, the given phenotype of a species mismatches the local temperature to the extent that it is too maladapted to be able to grow. Note the tradeoff between the width and height of the growth curves, with more warm-tolerant species having larger maximum growth at the cost of being viable for only a narrower range of temperatures[62,63]. **b** Temperature changes over time. After an initial establishment phase of 4000 years during which the pre-climate change community dynamics stabilize, temperatures start increasing at $t = 0$ for 300 years (vertical dotted line, indicating the end of climate change). Colors show temperature change at different locations along the spatial gradient, with warmer colors indicating lower latitudes. The magnitude and latitudinal dependence of the temperature change is based on region-specific predictions by 2100 CE, in combination with estimates giving an approximate increase by 2300 CE, for the IPCC intermediate emission scenario[27].

a single round of random mating restores the normal shape of the trait distribution, even if it is distorted by selection or migration. Second, the phenotypic variance is unchanged by these processes, with only the mean being affected[25] (we apply a reduction in genetic variance at very low population densities to prevent such species from evolving rapidly; see the Supplementary Information [SI], Section 3.4). Consequently, despite selection and the mixing of phenotypes from neighboring patches, each species retains a normally-shaped phenotypic distribution with the same phenotypic variance across all patches—but the mean temperature optimum may evolve locally and can therefore differ across patches (Fig. 1).

Species in our setup may either be resources or consumers. Their local dynamics are governed by the following processes. First, within each patch, we allow for migration to and from adjacent patches (changing both local population densities and also local adaptedness, due to the mixing of immigrant individuals with local ones). Second, each species' intrinsic rate of increase is temperature-dependent, influenced by how well their temperature optima match local temperatures (Fig. 2a). For consumers, metabolic loss and mortality always result in negative intrinsic growth, which must be compensated by sufficient consumption to maintain their populations. Third, there is a local competition between resource species, which can be thought of as exploitative competition for a set of shared substitutable lower-level resources[26]. Consumers, when present, compete only indirectly via their shared resource species. Fourth, each consumer has feeding links to five of the resource species (pending their presence in patches where the consumer is also present), which are randomly determined but always include the one resource which matches the consumer's initial mean temperature optimum. Feeding rates follow a Holling type II functional response. Consumers experience growth from consumption, and resource species experience loss due to being consumed.

Following the previous methodology, we derive our equations in the weak selection limit[22] (see also the Discussion). We have multiple selection forces acting on the different components of our model. Species respond to local climate (frequency-independent directional selection, unless a species is at the local environmental optimum), to consumers and resources

(frequency-dependent selection), and competitors (also frequency-dependent selection, possibly complicated by the temperature-dependence of the competition coefficients mediating frequency dependence). These different modes of selection do not depend on the parameterization of evolution and dispersal, which instead are used to adjust the relative importance of these processes.

Communities are initiated with 50 species per trophic level, subdividing the latitudinal gradient into 50 distinct patches going from pole to equator (results are qualitatively unchanged by increasing either the number of species or the number of patches; SI, Section 5.9–5.10). We assume that climate is symmetric around the equator; thus, only the pole-to-equator region needs to be modeled explicitly (SI, Section 3.5). The temperature increase is based on predictions from the IPCC intermediate emission scenario[27] and corresponds to predictions for the north pole to the equator. The modeled temperature increase is represented by annual averages and the increase is thus smooth. Species are initially equally spaced, and adapted to the centers of their ranges. We then integrate the model for 6500 years, with three main phases: (1) an establishment period from $t = -4000$ to $t = 0$ years, during which local temperatures are constant; (2) climate change, between $t = 0$ and $t = 300$ years, during which local temperatures increase in a latitude-specific way (Fig. 2b); and (3) the post-climate change period from $t = 300$ to $t = 2500$ years, where temperatures remain constant again at their elevated values.

To explore the influence and importance of dispersal, evolution, and interspecific interactions, we considered the fully factorial combination of high and low average dispersal rates, high and low average available genetic variance (determining the speed and extent of species' evolutionary responses), and four different ecological models. These were: (1) the baseline model with a single trophic level and constant, patch- and temperature-independent competition between species; (2) two trophic levels and constant competition; (3) single trophic level with temperature-dependent competition (where resource species compete more if they have similar temperature optima); and (4) two trophic levels as well as temperature-dependent competition. Trophic interactions can strongly influence diversity in a community, either by apparent competition[28] or by acting as

extra regulating agents boosting prey coexistence[29]. Temperature-dependent competition means that the strength of interaction between two phenotypes decreases with an increasing difference in their temperature optima. Importantly, while differences in temperature adaptation may influence competition, they do not influence trophic interactions.

The combination of high and low genetic variance and dispersal rates, and four model setups, gives a total of $2 \times 2 \times 4 = 16$ scenarios. For each of them, some parameters (competition coefficients, tradeoff parameters, genetic variances, dispersal rates, consumer attack rates, and handling times; SI, Section 6) were randomly drawn from pre-specified distributions. We, therefore, obtained 100 replicates for each of these 16 scenarios. While replicates differed in the precise identity of the species which survived or went extinct, they varied little in the overall patterns they produced.

We use the results from these numerical experiments to explore patterns of (1) local species diversity (alpha diversity), (2) regional trends, including species range breadths and turnover (beta diversity), (3) global (gamma) diversity, and global changes in community composition induced by climate change. In addition, we also calculated the interspecific community-wide trait lag (the difference between the community's density-weighted mean temperature optima and the current temperature) as a function of the community-wide weighted trait dispersion (centralized variance in species' density-weighted mean temperature optima; see Methods). The response capacity is the ability of the biotic community to close this trait lag over time[30] (SI, Section 4). Integrating trait lag through time[31] gives an overall measure of different communities' ability to cope with changing climate over this time period; furthermore, this measure is comparable across communities. The integrated trait lag summarizes, in a single functional metric, the performance and adaptability of a community over space and time. The reason it is related to performance is that species that on average live more often under temperatures closer to their optima (creating lower trait lags) will perform better than species whose temperature optima are far off from local conditions in space and/or time. Thus, a lower trait lag (higher response capacity) may also be related to other ecosystem functions, such as better carbon uptake which in turn has the potential to feedback to global temperatures[32].

**Overview of results**. We use our framework to explore the effect of species interactions on local, regional, and global biodiversity patterns, under various degrees of dispersal and available genetic variance. For simplicity, we focus on the dynamics of the resource species, which are present in all scenarios. Results for consumers, when present, are in the SI (Section 5.8). First, we display a snapshot of species' movement across the landscape with time; before, during, and after climate change. Then we proceed with analyzing local patterns, followed by regional trends, and finally, global trends.

Snapshots from the time series of species' range distributions reveal useful information about species' movement and coexistence (Fig. 3). Regardless of model setup and parameterization, there is a northward shift in species' ranges: tropical species expand into temperate regions and temperate species into polar regions. This is accompanied by a visible decline in the number of species globally, with the northernmost species affected most. The models do differ in the predicted degree of range overlap: trophic interactions and temperature-dependent competition both lead to broadly overlapping ranges, enhancing local coexistence (the overlap in spatial distribution is particularly pronounced with high available genetic variance). Without these interactions,

species ranges overlap to a substantially lower degree, diminishing local diversity. Below we investigate whether these patterns, observed for a single realization of the dynamics for each scenario, play out more generally as well.

**Local trends**. Trophic interactions and temperature-dependent competition indeed result in elevated local species richness levels (Fig. 4). The fostering of local coexistence by trophic interactions and temperature-dependent competition is in line with general ecological expectations. Predation pressure can enhance diversity by providing additional mechanisms of density regulation and thus prey coexistence through predator partitioning[28,29]. In turn, temperature-dependent competition means species can reduce interspecific competition by evolving locally suboptimal mean temperature optima[22], compared with the baseline model's fixed competition coefficients. Hence with temperature-dependent competition, the advantages of being sufficiently different from other locally present species can outweigh the disadvantages of being somewhat maladapted to the local temperatures. If competition is not temperature-dependent, interspecific competition is at a fixed level independent of the temperature optima of each species. An important question is how local diversity is affected when the two processes act simultaneously. In fact, any synergy between their effects is very weak, and is even slightly negative when both the available genetic variance and dispersal abilities are high (Fig. 4, top row).

**Regional trends**. We see a strong tendency for poleward movement of species when looking at the altered distributions of species over the spatial landscape (Fig. 3). Indeed, looking at the effects of climate change on the fraction of patches occupied by species over the landscape reveals that initially cold-adapted species lose suitable habitat during climate change, and even afterwards (Fig. 5). For the northernmost species, this always eventuate to the point where all habitat is lost, resulting in their extinction. This pattern holds universally in every model setup and parameterization. Only initially warm-adapted species can expand their ranges, and even they only do so under highly restrictive conditions, requiring both good dispersal ability and available genetic variance as well as consumer pressure (Fig. 5, top row, second and third panel).

One can also look at larger regional changes in species richness, dividing the landscape into three equal parts: the top third (polar region), the middle third (temperate region), and the bottom third (tropical region). Region-wise exploration of changes in species richness (Fig. 6) shows that the species richness of the polar region is highly volatile. It often experiences the greatest losses; however, with high dispersal ability and temperature-dependent competition, the regional richness can remain substantial and even increase compared to its starting level (Fig. 6, first and third rows, last two columns). Of course, change in regional species richness is a result of species dispersing to new patches and regions as well as of local extinctions. Since the initially most cold-adapted species lose their habitat and go extinct, altered regional species richness is connected to having altered community compositions along the spatial gradient. All regions experience turnover in species composition (SI, Section 5.1), but in general, the polar region experiences the largest turnover, where the final communities are at least 50% and sometimes more than 80% dissimilar to the community state right before the onset of climate change—a result in agreement with previous studies as well[7,33].

**Global trends**. Hence, the identity of the species undergoing global extinction is not random, but strongly biased towards

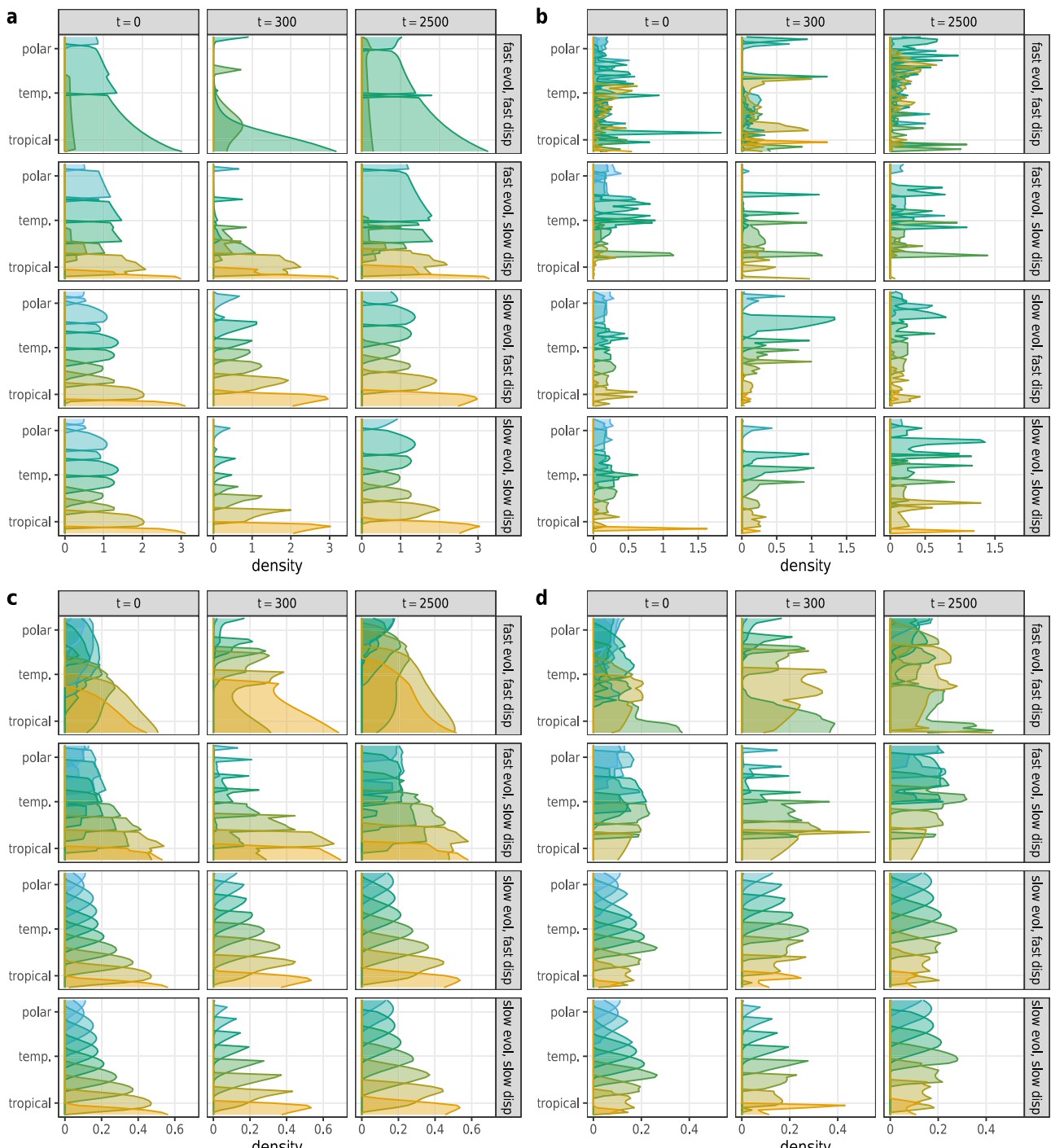

**Fig. 3 Species' range shift through time, along a latitudinal gradient ranging from polar to tropical climates (ordinate).** Species distributions are shown by colored curves, with the height of each curve representing local density in a single replicate (abscissa; note the different scales in the panels), with the color indicating the species' initial (i.e., at $t = 0$) temperature adaptation. The model was run with only 10 species, for better visibility. The color of each species indicates its temperature adaptation at the start of the climate change period, with warmer colors belonging to species with a higher temperature optimum associated with higher latitudes. Rows correspond to a specific combination of genetic variance and dispersal ability of species, columns show species densities at different times ($t = 0$ start of climate change, $t = 300$ end of climate change, $t = 2500$ end of simulations). Each panel corresponds to a different model setup; **a** the baseline model, **b** an added trophic level of consumers, **c** temperature-dependent competition coefficients, and **d** the combined influence of consumers and temperature-dependent competition.

initially cold-adapted species. On a global scale, these extinctions cause decreased richness, and the model predicts large global biodiversity losses for all scenarios (Fig. 6). These continue during the post-climate change period with stable temperatures, indicating a substantial extinction debt which has been previously demonstrated[34]. Temperature-dependent competition reduces the number of global losses compared to the baseline and trophic models.

A further elucidating global pattern is revealed by analyzing the relationship between the time-integrated temperature trait lag

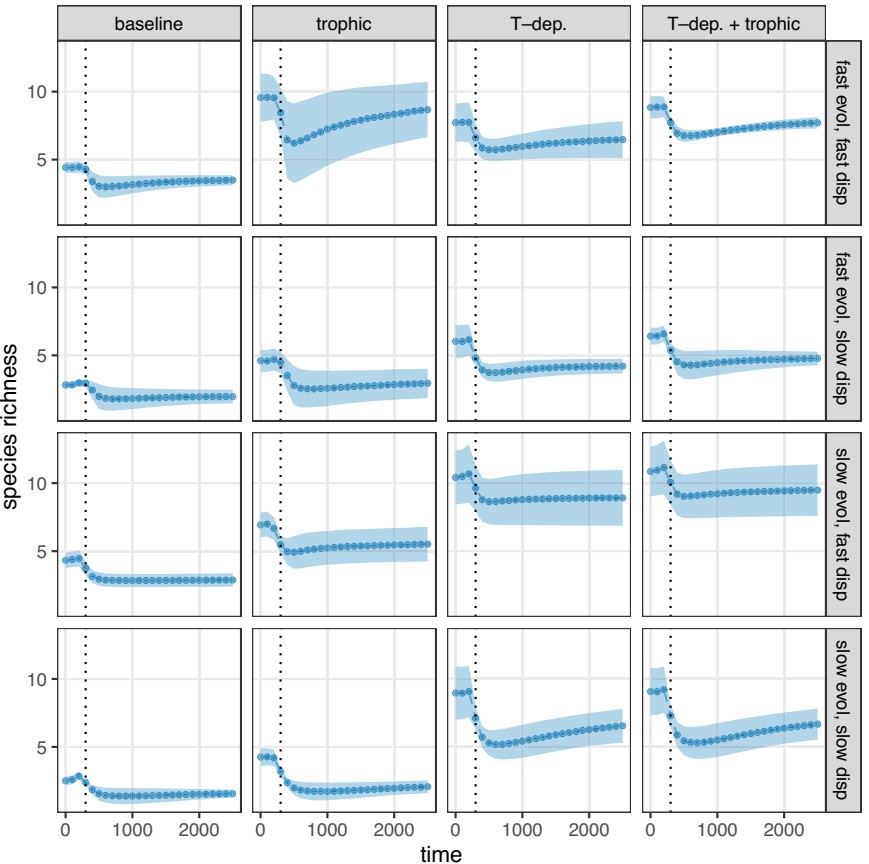

**Fig. 4 Local species richness of communities over time, from the start of climate change to the end of the simulation, averaged over replicates.** Values are given in 100-year steps. At each point in time, the figure shows the mean number of species per patch over the landscape (points) and their standard deviation (shaded region, extending one standard deviation both up- and downwards from the mean). Panel rows show different parameterizations (all four combinations of high and low genetic variance and dispersal ability); columns represent various model setups (the baseline model; an added trophic level of consumers; temperature-dependent competition coefficients; and the combined influence of consumers and temperature-dependent competition). Dotted vertical lines indicate the time at which climate change ends.

and community-wide trait dispersion (Fig. 7). There is an overall negative correlation between the two, but more importantly, within each scenario (unique combination of model and parameterization) a negative relationship is evident. Furthermore, the slopes are very similar: the main difference between scenarios is in their mean trait lag and trait dispersion values (note that the panels do not share axis value ranges). The negative trend reveals the positive effect of more varied temperature tolerance strategies among the species on the community's ability to respond to climate change. This is analogous to Fisher's fundamental theorem[35], stating that the speed of the evolution of fitness $r$ is proportional to its variance: $dr/dt \sim var(r)$. More concretely, this relationship is also predicted by trait-driver theory, a mathematical framework that focuses explicitly on linking spatiotemporal variation in environmental drivers to the resulting trait distributions[30]. Communities generated by different models reveal differences in the magnitude of this relationship: trait dispersion is much higher in models with temperature-dependent competition (essentially, niche differentiation with respect to temperature), resulting in lower trait lag. The temperature-dependent competition also separates communities based on their spatial dispersal ability, with faster dispersal corresponding to greater trait dispersion and thus lower trait lag. Interestingly, trophic interactions tend to erode the relationship between trait lag and trait dispersion slightly ($R^2$ values are lower in communities with trophic interactions, both with and without temperature-dependent competition). We have additionally

explored the relationship between species richness and trait dispersion, finding a positive relationship between the two (SI, Section 4.1).

## Discussion

**General modeling considerations.** This work introduced a modeling framework combining dispersal, evolutionary dynamics, and ecological interactions in a way that is tractable, easy to implement, fast to execute on a computer, and can handle ecological interactions of realistic complexity without simultaneously breaking other aspects of the approach. Individual-based models[6], for instance, do in principle allow one to include arbitrary levels of complexity, but tend to be computationally expensive. Other models yield detailed projections of individual species and their genetic structure but ignore species interactions altogether[36]. An intermediate approach is based on quantitative genetics, which takes species interactions into account and yields a description of species' genetic structure that is sufficiently simplified to be tractable. Earlier models in this spirit[7,13,37] were built on coupled partial differential equations. While the theory behind such models is highly elegant, coupled nonlinear partial differential equations are notoriously difficult to implement in a way that is numerically stable, yields accurate results, and does not require unacceptably long run-times—notably, naive discretization schemes often do not work well. Unfortunately, despite persistent warnings about these problems[38], such naive solution schemes still prevail in the literature.

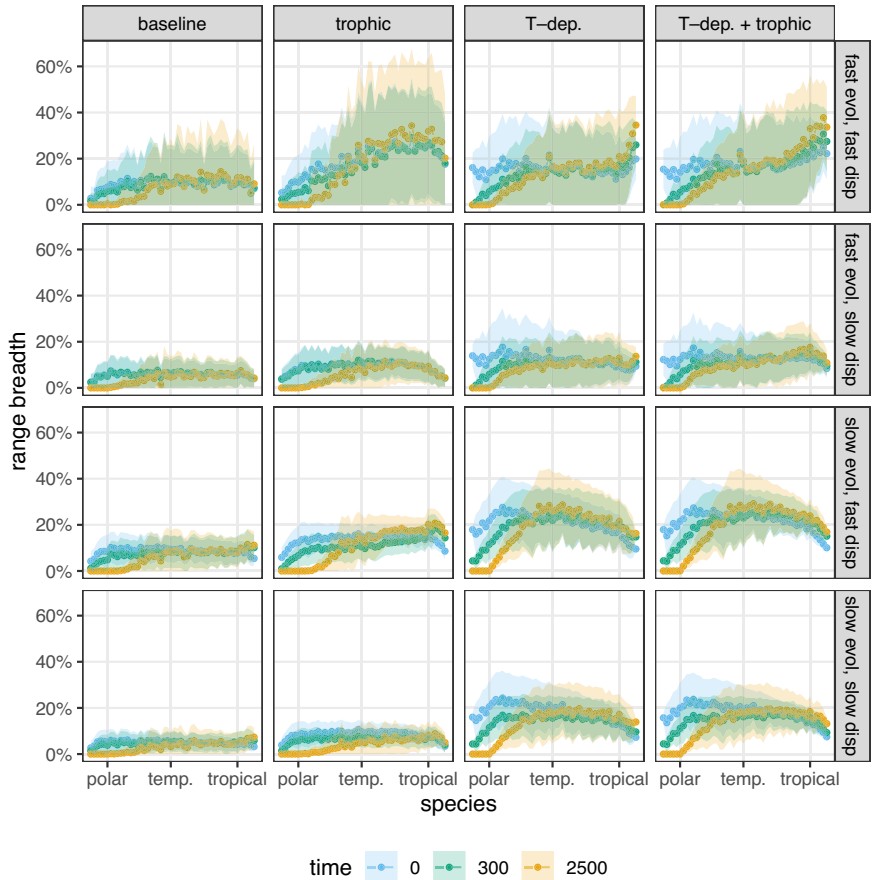

**Fig. 5 Range breadth of each species expressed as the percentage of the whole landscape they occupy (ordinate) at three different time stamps (colors).** The mean (points) and plus/minus one standard deviation range (colored bands) are shown over replicates. Numbers along the abscissa represent species, with initially more warm-adapted species corresponding to higher values. The range breadth of each species is shown at three time stamps: at the start of climate change ($t = 0$, blue), the end of climate change ($t = 300$, green), and at the end of our simulations ($t = 2500$, yellow). Panel layout as in Fig. 4.

We circumvented this problem by building, from first principles, a different framework for spatial eco-evolutionary dynamics. Within a single patch, it is based on a quantitative genetic recursion model[19,22]. Spatial locations are discretized from the outset, therefore the approach is built on ordinary differential equations alone. As a consequence, it executes fast even with substantial model complexity: on an ordinary desktop computer, a single run for 6500 years with both trophic interactions and temperature-dependent competition, with 50 species on both trophic levels and 50 habitat patches (for $100 \times 50 \times 2 = 10{,}000$ state variables; the factor of 2 is because both the density and trait mean of each species may change) finished in around 3 min. While this is fast, emerging methods such as Universal Differential Equations, which combine traditional integration with machine learning, hold the promise of a many-fold increase in the speed of computation in the near future[39]. Incorporation of further complexity into our model is straightforward: complex food webs and spatial structure, or further trait variables under selection (e.g., having both temperature optimum and body size evolve, the latter dictating the type of prey a species can consume[40]), can all be implemented. An important future extension would be to use an improved climate model with annual temperature fluctuations, instead of our smooth increase based on annual means. Annual extreme weather events are expected to become more common[41]. Under such circumstances, Allee effects might mean more frequent extinctions than predicted from our current model,

because species hit by such events might not be able to recover. On the other hand, annual temperature cycles could induce storage effects or relative nonlinearities[42,43], which in combination with our already incorporated spatial variation could promote coexistence through joint spatial and temporal variation[44].

We derive our equations using the idealizations of additive quantitative genetics and the weak selection limit[22]. Both have their drawbacks. The first assumes that all genetic variation is additive—genes and alleles at different loci do not interact. This ignores the fact that genes are part of a complex regulatory network in which interactions such as dominance, epistasis, and pleiotropy are bound to emerge. While purely additive quantitative genetics can be a good starting point for understanding the effects of selection[45,46], it remains an approximation. In turn, the weak selection limit assumes that selection is not so strong as to prevent one from writing otherwise discrete-time dynamics in the continuous-time limit (SI, Section 1). In fact, from a practical point of view, this limit can actually allow for quite a strong selection. For this, however, one must assume very large population sizes so genetic and ecological drift do not overpower selection. The rule of thumb is that effective population size times the selection differential must exceed one[47]. This is obviously true if populations are so large that they can be modeled as continuous variables, but in reality, they are finite, and the weak selection assumption could potentially yield effects which we neglect. For example, a new immigrant at a habitat patch will naturally have a

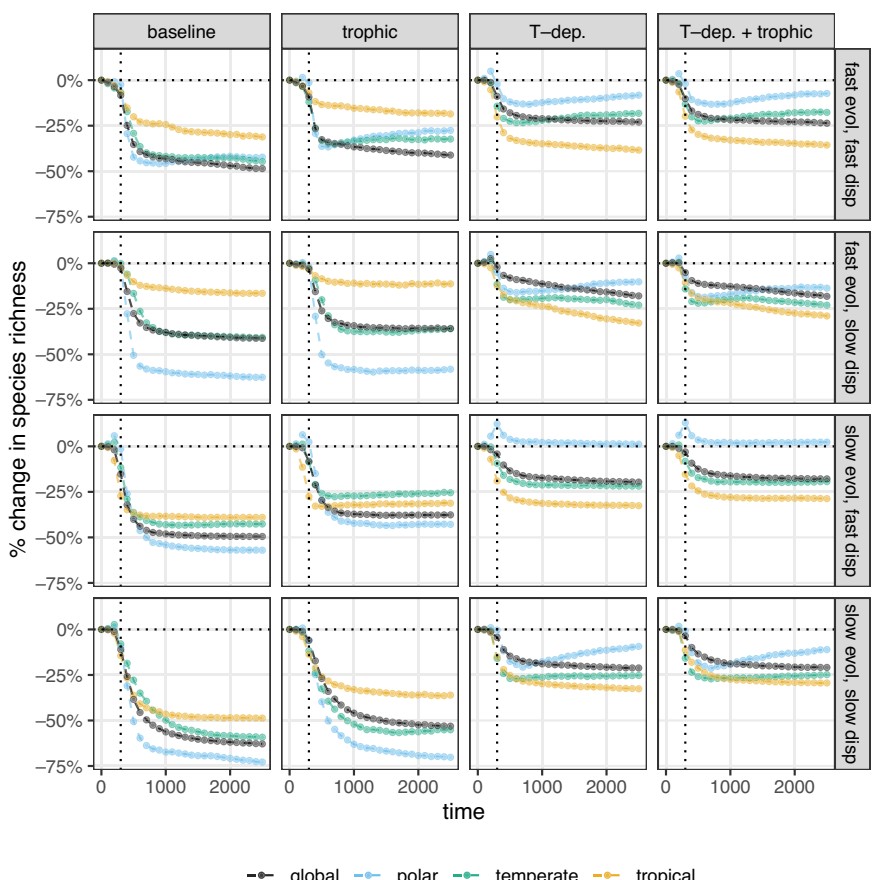

**Fig. 6 Relative change in global species richness from the community state at the onset of climate change (ordinate) over time (abscissa), averaged over replicates and given in 100-year steps (points).** Black points correspond to species richness over the whole landscape; the blue points to richness in the top third of all patches (the polar region), green points to the middle third (temperate region), and yellow points to the last third (tropical region). Panel layout as in Fig. 4; dotted horizontal lines highlight the point of no net change in global species richness.

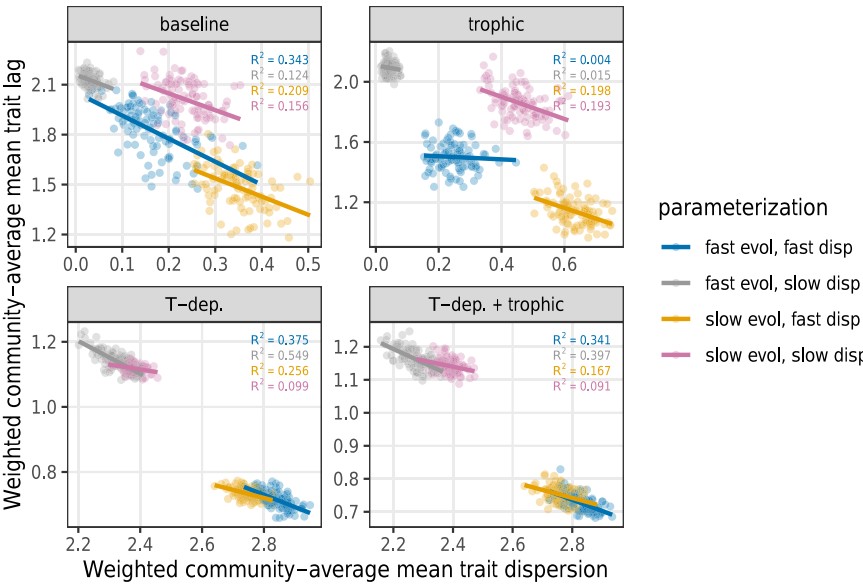

**Fig. 7 The ability of communities in four different models (panels) to track local climatic conditions (ordinate), against observed variation in traits within those communities (abscissa).** Larger values along the ordinate indicate that species' temperature optima are lagging behind local temperatures, meaning a low ability of communities to track local climate conditions. Both quantities are averaged over the landscape and time from the beginning to the end of the climate change period, yielding a single number for every community (points). The greater the average local diversity of mean temperature optima in a community, the closer it is able to match the prevailing temperature conditions. Species' dispersal ability and available genetic variance (colors) are clustered along this relationship.

low population size and might not be able to establish even if it has higher fitness. Similarly, a slightly deleterious type will never spread in our approximation, while it might in reality, as known from the nearly neutral theory[48]. Although Akashi et al.[48] show that weak selection can often explain similar patterns of genome variation as the nearly neutral theory, a rigorous incorporation of the consequences of population finiteness in our model is still in the future.

There is one other important thing our model currently cannot do. Since trait distributions are assumed to be normal with constant variance, a species cannot split into two daughter lineages in response to disruptive selection, as this would require the trait distribution to become gradually more and more bimodal. As such, our model ignores speciation, which may turn out to be an important process in regions that become species-impoverished following climate change. In sexual populations, speciation can occur when the trait is a magic trait, which jointly drives competitive interactions but also assortative mating between similar phenotypes[49,50]. Here we strictly assume that temperature tolerance is not involved in mate choice. This, however, is an oversimplification because magic traits may in fact be very common in nature[51]. And magic traits may not even be necessary, since pseudo-magic traits (with two tightly linked loci, one under divergent selection and the other acting as a mating cue) can also promote speciation[52]. There are also non-ecological (e.g., mutation-order) speciation mechanisms that could play a role[53]. New species emerging by speciation could possibly mitigate the decrease of species we currently observe. But our model, in its current form, cannot incorporate these mechanisms.

**The role of species interactions**. Using our framework, we demonstrate that biotic factors such as trophic interactions and temperature-dependent competition are important in shaping species' eco-evolutionary response to climate change—in fact, they can be as influential as the ability of species to adapt to new local climates or to disperse to new habitats. With trophic interactions and temperature-dependent competition, species have broader ranges and coexist to a higher degree, in comparison to the baseline model without the aforementioned dynamics. In addition, temperature-dependent competition significantly reduces global species loss. With constant competition as in our baseline scenario, competitive exclusions occur to a higher extent, a result in line with van Eldijk et al.[10], showing that evolutionary rescue of one species leads to a competitive exclusion and extinction of another species. The importance of biotic interactions for shaping species' response to climate change is well-known[8,10,15,16]. Our work complements these studies by further demonstrating the significance of biotic interactions in an eco-evolutionary setting as well. The mechanisms behind this are predator-mediated coexistence[28,29] (in the case of trophic interactions), and reduced interspecific competition with increasing trait distance[22]. Note that this last mechanism is not guaranteed to promote diversity, since the level of difference in mean temperature optima required for significant reductions in competition might mean that species have local growth rates that are too suboptimal for persistence. Thus, the ability of this mechanism to boost diversity depends on whether species are able to tolerate suboptimal climates sufficiently to avoid local competition.

There are interaction types we do not consider in this version of our framework. Similar to our modeling of competition, one could have temperature-dependent mutualism; i.e., the strength of the mutualistic benefit between two phenotypes is a function of the distance of their temperature optima. This process could potentially bind mutualistic species to a common fate[54] and thus accelerate the effects of climate change. Indeed, Northfield, Ives[55]

showed that with non-conflicting evolution of mutualistic interactions, the effects of climate change are enhanced, and the dynamics are destabilized. Our model is extensible to incorporate other types of interactions and structures (e.g., modular or nested ones, either of trophic or mutualistic interactions). These are important future problems to address in the context of eco-evolutionary responses to climate change.

**The role of dispersal and genetic variance**. Besides the importance of biotic interactions affecting species' persistence and distribution under climate change, we also show that their dispersal ability and available genetic variance (i.e., capacity to respond to selective pressures swiftly) influence their responses. When local conditions change and temperatures increase, species become increasingly maladapted at their initial locations and pre-adapted to temperatures at higher latitudes, driving a northward movement. Dispersal is therefore suggested as a mechanism that provides spatial insurance to species[56,57], mitigating the negative impacts of climate change. However, a northward movement of initially warm-adapted species comes at the expense of the species located in the coldest regions which cannot disperse further[33], a consequence of dispersal that has been shown in the previous studies[7]. One might think that combining good dispersal ability with large genetic variance should temper this problem by allowing the northernmost species to adapt locally, and thus alleviate the negative impacts of increased temperatures better than each of these processes on their own. This expectation is also consistent with recent projections based on empirical data[58]. However, the projected extinctions, considering both dispersal and species' ability to adapt, have been obtained without explicitly considering species interactions[58]. We show that large genetic variance combined with good dispersal ability result in a global biodiversity loss similar to when both dispersal ability and evolutionary rate are low. The reason, again, has to do with species interactions: the ability of individual species to disperse and adapt to new local conditions is of no use if they are prevented by other species from reaching the new locations. Similarly, cold-adapted species may be able to sustain in their current location with large genetic variance, but get outcompeted by the arrival of better adapted migrating species. The negative interaction between high dispersal and fast adaptation under climate change has also been demonstrated by Thompson and Fronhofer[6]. However, in our case, we show that temperature-dependent competition reduces some of the negative impacts by allowing more local coexistence, albeit at the cost of reduced local growth rates.

**Trait lag and trait dispersion**. We show that models in which communities are able to maintain high biodiversity after altered climatic conditions will in general also have high trait diversity and low trait lag. This particularly occurs when species have temperature-dependent competition, allowing species to exploit different microhabitats within the same patch. High trait diversity results in high response capacity of the community to climate change and thus a lower overall trait lag. Species richness and trait dispersion can potentially be statistically correlated, as often found in biodiversity and ecosystem functioning studies[59]—although in our simulations this positive relationship holds only for the aggregated data as a whole, not necessarily within each individual model parameterization (SI, Section 4.1).

The trait in our study—the temperature optimum of each species—can also be regarded as a functional trait explaining how species share a resource. In our case, this is expressed as species' exploration of habitats with suitable temperatures. High functional trait diversity has then been shown to be important for sustaining multiple ecosystem functions simultaneously, since

coexisting species can exploit different resources and microhabitats[60,61]. It is encouraging for the general predictability of biotic climate impact models that the resulting trait dispersion in temperature-related traits strongly correlates with the ability of the community to cope with climate change. This can justify putting the focus on processes that can sustain local community-wide trait dispersion, providing an argument for general biodiversity-enhancing measures such as preserving habitat heterogeneity, maintaining populations of keystone species, and for constructing dispersal corridors.

The focus on trait dispersion has an important and complementary implication for traditional conservation strategies of more biodiversity is better. It simplifies the identification of strategies that underpin the maintenance of trait variation of a particular trait and thus a particular environmental driver that is of concern. For example, ensuring connectivity to habitats with higher mean temperature or temperature variation can promote an influx of species or genotypes that can cope with an increasing trend in temperature by maintaining the local trait variation of temperature optima. Local management strategies can target microhabitats that have south-facing sheltered microclimates to promote islands of environmental conditions that reflect possible future scenarios.

**Conclusions.** Biological communities are affected by many factors, ecological as well as evolutionary, which influence their response to climate change. Our framework demonstrates the importance of including relevant biological processes for predicting large-scale consequences of climate change on global and local biodiversity. Realistic mechanisms such as species interactions over multiple trophic levels and temperature-dependent competition, as well as particular combinations of dispersal and available genetic variance, can alleviate some of the negative impacts of climate change, showing potential ways for ecological communities to adjust to altered climatic conditions. Despite this, the negative impact of climate change on ecological communities is severe, with numerous global extinctions and effects that are manifested long after the climate has again stabilized.

## Methods

We consider a chain of $L$ evenly spaced patches along a latitudinal gradient, where patches 1 and $L$ correspond to the north pole and equator, respectively. The temperature $T^k(t)$ in patch $k$ at time $t$ is given by

$$T^k(t) = \underbrace{\left(T_{\min} + (T_{\max} - T_{\min})\frac{k}{L}\right)}_{\text{initial temperature profile}} + \underbrace{\left(C_{\max} + (C_{\min} - C_{\max})\frac{k}{L}\right)}_{\text{total temperature change}} \underbrace{Q(t/t_E)}_{\% \text{ change at time } t}. \tag{1}$$

$T_{\min}$ and $T_{\max}$ are the initial polar and equatorial temperatures; $C_{\max}$ and $C_{\min}$ are the corresponding temperature increases after $t_E = 300$ years, based on the IPCC intermediate emission scenario[27]. The period from $t = -4000$ to $t = 0$ is an establishment time preceding climate change. $Q(\tau)$ describes the sigmoidal temperature increase in time: $Q(\tau)$ equals 0 for $\tau < 0$, 1 for $\tau > 1$, and $10\tau^3 - 15\tau^4 + 6\tau^5$ otherwise. Figure 2b depicts the resulting temperature change profile.

Combining quantitative genetics with dispersal across the $L$ patches, we track the population density and mean temperature optimum of $S$ species. Let $N_i^k$ be the density and $\mu_i^k$ the mean temperature optimum of species $i$ in patch $k$ (subscripts denote species; superscripts patches). The governing equations then read

$$\frac{dN_i^k}{dt} = \underbrace{N_i^k \int r_i^k(z)p_i^k(z)dz}_{\text{local population growth}} + \underbrace{\sum_{l=1}^{L} m_i^{kl} N_i^l}_{\text{immigration}} - \underbrace{\sum_{l=1}^{L} m_i^{lk} N_i^k}_{\text{emigration}}, \tag{2}$$

$$\frac{d\mu_i^k}{dt} = \underbrace{h_i^2 \int (z - \mu_i^k)r_i^k(z)p_i^k(z)dz}_{\text{local selection}} + \underbrace{h_i^2 \sum_{l=1}^{L} m_i^{kl} \frac{N_i^l}{N_i^k}(\mu_i^l - \mu_i^k)}_{\text{trait change from immigration}} \tag{3}$$

(SI, Section 1), where $t$ is time, $r_i^k(z)$ the per capita growth rate of species $i$'s phenotype $z$ in patch $k$, $p_i^k(z)$ species $i$'s temperature optimum distribution in patch

$k$ (which is normal with patch-dependent mean $\mu_i^k$ and patch-independent variance $\sigma_i^2$), $h_i^2$ the heritability of species $i$'s temperature optimum, and $m_i^{kl}$ the migration rate of species $i$ from patch $l$ to $k$. The per capita growth rates $r_i^k(z)$ read

$$r_i^k(z) = r_{0,i}^k(z) - \sum_{j=1}^{S} N_j^k \int a_{ij}^k(z,z')p_j^k(z') \, dz' + \sum_{j=1}^{S} \epsilon_i F_{ij}^k - \sum_{j=1}^{S} N_j^k F_{ji}^k/N_i^k. \tag{4}$$

Here

$$r_{0,i}^k(z) = \left(\frac{\varrho_i}{b_w - a_w\mu_i^k}\right)\exp\left(-\frac{(T^k - z)^2}{2(b_w - a_w\mu_i^k)^2}\right) - \kappa_i \tag{5}$$

is the intrinsic growth of species $i$'s phenotype $z$ in patch $k$. The constants $\varrho_i$, $b_w$, and $a_w$ modulate a tradeoff between maximum growth and tolerance range[62,63] (Fig. 2a), $\kappa_i$ is a mortality rate, and $T^k$ is the current local temperature in patch $k$. In turn, $a_{ij}^k(z,z')$ is the competition coefficient between species $i$'s phenotype $z$ and species $j$'s phenotype $z'$ in patch $k$. We either assume constant, patch- and phenotype-independent coefficients $a_{ij}$, or ones which decline with increasing trait differentiation according to

$$a_{ij}^k(z,z') = \exp\left(-\frac{(z-z')^2}{\eta^2}\right) \tag{6}$$

(temperature-dependent competition), where $\eta$ is the competition width. The parameter $\epsilon_i$ in Eq. (4) is species $i$'s resource conversion efficiency, and $F_{ij}^k$ is the feeding rate of species $i$ on $j$ in patch $k$:

$$F_{ij}^k = \frac{q_i W_{ij}\omega_{ij}N_j^k}{1 + q_i H_i \sum_{s=1}^{S} W_{is}\omega_{is}N_s^k}, \tag{7}$$

where $q_i$ is species $i$'s attack rate, $W_{ij}$ is the adjacency matrix of the feeding network ($W_{ij} = 1$ if $i$ eats $j$ and 0 otherwise), $\omega_{ij}$ is the proportion of effort of $i$ on $j$, and $H_i$ is species $i$'s handling time. When adding a second trophic level, the number of species on the new level is equal to that at the lower level, and each consumer is linked with five resource species in a bipartite feeding network (SI, Section 3.3).

We numerically integrated 100 replicates for each of 16 scenarios, made up of the fully factorial combinations of:

- The average dispersal rate between adjacent patches, which was either high (100 m/yr) or low (0.01 m/yr).
- The mean genetic variance per species, also either high ($10^{-1}$°C$^2$) or low ($10^{-3}$°C$^2$).
- The model setup, which was one of the following:

1. One trophic level and constant competition coefficients, $a_{ij}^k(z,z') = a_{ij}$.
2. Two trophic levels and constant competition coefficients.
3. One trophic level and competition coefficients given by Eq. (6).
4. Two trophic levels and competition coefficients given by Eq. (6).

For each replicate, all other parameters are assigned based on Section 6 in the SI. Numerical integration of the system starts at $t_0 = -4000$ years, with initial conditions

$$\mu_i^k(t_0) = (T_{\max} - T_{\min})\frac{i}{S} + T_{\min} \tag{8}$$

and

$$N_i^k(t_0) = \exp\left(-\frac{(\mu_i^k(t_0) - T^k(0))^2}{8}\right) \tag{9}$$

(SI, Section 3.7), and terminates at $t = 2500$ years.

The community-average trait dispersion $\mathcal{V}^k$ of the local community in patch $k$ is the density-weighted variance of species' mean temperature optima:

$$\mathcal{V}^k = \sum_{i=1}^{S} n_i^k\left(\mu_i^k - \bar{\mu}^k\right)^2, \tag{10}$$

where $n_i^k = N_i^k/\sum_{j=1}^{S} N_j^k$ is the relative density of species $i$ in patch $k$, and $\bar{\mu}^k = \sum_{i=1}^{S} n_i^k\mu_i^k$ is the community-weighted average of species' temperature optima in patch $k$. In turn, the community-average trait lag $\mathcal{A}^k$ in patch $k$ is defined as the difference between the local temperature $T^k$ and the local community-weighted mean trait $\bar{\mu}^k$:

$$\mathcal{A}^k = T^k - \bar{\mu}^k. \tag{11}$$

In Fig. 7, these quantities are averaged over all patches of the landscape and over time, from the beginning to the end of climate change. These averages are taken separately for each of the 1600 model realizations (16 scenarios, with 100 replicates each).

## Data availability

The computer-generated data of this study has been deposited and can be downloaded from https://zenodo.org/record/5060300[64].

## Code availability

Computer code for implementing our model and replicating our results can be found at https://zenodo.org/record/5060300[64].

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

## Acknowledgements

We thank Priyanga Amarasekare and Peter Münger for discussions. This research was funded by the Swedish Research Council (grant FORMAS 2015-01262 to A.E., and grant VR 2017-05245 to G.B.).

## Author contributions

A.Å., B.E., and A.C. conceived of the initial study. G.B. wrote the supplement and developed the temperature-tolerance concept. J.N. contributed the trait-lag concept. A.Å. and G.B. performed data simulations and analysis. A.Å, G.B., and A.E. wrote the paper. All authors made significant edits to the final version of the manuscript.

## Funding

## Competing interests

The authors declare no competing interests.
