## [Peer Review File · Nature Communications]

REVIEWER COMMENTS

Reviewer #1 (Remarks to the Author):

The authors study how different models of ecological communities and varying genetic diversity and dispersal ability influence community response to sustained environmental change.

Overall I think the authors have chosen an exciting line of research and questions. I think there are still many exciting and important lessons to be learned about ecological and evolutionary processes that underly how biodiversity at multiple levels will respond to environmental change. The authors have made important advances over previous work by increasing the realism or complexity of their evolutionary community models. However, I am concerned that despite this advance, the paper lacks a coherent central message with key lessons learned. Rather, I find that paper feels a bit sprawling. Admittedly it is challenging to find a pithy lesson in the tangled bank of complex community eco-evo responses to environmental change. Nevertheless, I think the exercise of communicating these findings would be greatly improved by developing a clear, well-connected thread of lessons learned.

The Introduction doesn't set the stage with a strong conceptual framework, it reads more like a laundry list (without strong organization) of potentially important processes and modeling issues to consider. "include how species interactions themselves change because of increasing temperatures" (L44) – this idea may be what the authors want to build their organization around. This issue comes up as well with the response variables of interest – is there a way to succinctly group and organize these?

This manuscript does not benefit from a Results section before the Methods. It is hard to interpret the differences in the different models without some more in depth description of those models, that may naturally lead to hypotheses.

I also have trouble with the way in which results are presented. They are generally abstracted to a high level and then described in very broad terms, and collapsing down a lot of parameter variation down to a few categories of process and pattern. There are benefits to this approach, to help avoid getting lost in too much detail, but I find that without some examples of in depth discussion of specific results, it is harder for the reader to firmly grasp what underlies the patterns identified here.

I disagree with this interpretation of trait lag patterns, where the authors seem to suggest there is no predictable effect of genetic variance or dispersal on trait lag: "Interestingly, the same clustering of results is absent with respect to model parameterization in terms of available genetic variance and dispersal ability." (L185). It is clear from Fig 6 that there are effects of genetic variance and dispersal. For example, nearly all the '+' fall above the trend line, indicating greater lag than expected when there is low genetic variance and

low dispersal ability, regardless of the type of community model. Circle symbols fall nearly all below the trend line, indicating less lag than expected for high genetic variance and high dispersal.

I am confused a bit by how with temperature dependent competition species “coexist to a higher degree” (L231) yet there is also lower trait diversity within communities (Fig 6). How does more coexistence and species diversity also lead to lower trait diversity?

The last two sentences of the Abstract should be worded more clearly. They’re a bit confusing.

L45 – I’d reword to say ‘has been little considered’ or something like that, as opposed to the strong priority claim.

L141 – the point here is unclear – from the wording, I cannot tell whether there is a scenario where cold adapted species expand ranges or whether there is no such scenario.

L179 – perhaps worth noting that this hypothesis is directly analogous to Fisher’s fundamental theorem

Figure 1 would benefit from some more labeling on the figure itself – especially the right panel. Labels on the curves shown could help. It is also a bit confusing that the species shown here are maladapted to the local temperature.

Figure 4 would benefit from a label on the x-axis or in the margin that indicates which species are polar and which tropical.

Reviewer #2 (Remarks to the Author):

Recent studies have demonstrated the importance of rapid evolution, species interactions, and dispersal for understanding the effects of climate change on biodiversity. This manuscript constructed a simulation model where discrete patches are connected by dispersal, and species evolve in response to climate change with competition and predation. The authors found that predation and temperature-dependent competition could mitigate the negative effect of climate change. Furthermore, large trait variance within the community could reduce the lag between the community mean trait and environmental optimum. Because of the importance of predicting and understanding eco-evolutionary dynamics under climate change, this study is of interest to many people in this research area. However, I feel the manuscript has some weaknesses to be addressed.

Major comments:

1. What are surprising results of this study? It is well known that specialist predators can promote prey coexistence (as kill-the-winner hypothesis), and larger trait variation allows rapid responses to changing

environments (e.g., Kopp & Matuszewski 2014 *Evol. Appl.*). And the mitigating effect of predation and temperature-dependent competition on extinction may be examined without considering dispersal. If this paper is to construct a new modeling framework (by combining quantitative genetic modeling and discrete spatial structure instead of using PDEs with continuous space), it may be better to publish in, for example, *Methods in Ecology & Evolution*.

2. It is not clear how temperature affect competition in Abstract and Introduction. Based on Equation 6 and the explanations in L321-323, readers who are familiar with math can see how it works, but they may still wonder why the competition coefficient, a , is not a function of temperature, T . Please add careful explanations to Abstract and Introduction as this is one of the most important point of this study. Although it seems that the authors assumed that temperature-dependent competition reduces the strength of interspecific competition (L153-154), it may be possible to see stronger interspecific competition due to temperature-dependence, right?

3. As shown in Figure 1, dispersal from neighboring patches can result in multimodal distribution of species traits, but the authors explained that they assumed unimodal, normal trait distributions (L221-226). How did the authors use the quantitative genetics framework to deal with this? Please add explanations to Introduction.

4. Figures 3, 4, and 5 have 16 small figures and it is sometimes unclear which figures the authors are talking about. It will be great if the authors can add characters (a , b , c ,...) to the small figures so that readers can understand which is which.

5. Although the authors concluded that predation can reduce extinction events, it is not clear how the generality of predators affects this outcome. While specialist predators promote negative density-dependence in community dynamics and cause coexistence, generalist predators may promote apparent competition and make coexistence difficult. It will be great if the authors can change the generality of predators to see the robustness of their results.

6. The authors considered competition and predation as species interactions, but how do mutualistic interactions change eco-evolutionary dynamics under climate change? Northfield & Ives (2013) *PLOS Biol.* showed that coevolution with mutualism can destabilize dynamics whereas coevolution with competition and predation tend to stabilize dynamics. It will be great to discuss the perspectives in Discussion.

7. It would be great if the authors focus on a single patch and examine whether the same patterns reported in previous studies on evolutionary rescue with interspecific interactions occur or not. For example, competition (de Mazancourt et al. 2008 *Ecol. Lett.*, Osmond & de Mazancourt 2013 *Phil. Trans. R. Soc. B*) and predation (Osmond et al. 2017 *Am. Nat.*, Cortez & Yamamichi 2019 *Ecology*) may promote or prevent extinction depending on model assumptions and parameters.

Minor comments:

P1

Title: It is hard to know the results of this study from the current title. What about "Specialist predation and temperature-dependent competition mitigate the negative impact of climate change on biodiversity" or something like this?

Authors: What is the affiliation of Bo Ebenman?

L5: What is the difference between species- and community-level dynamics? Did the authors mean population- and community-level dynamics?

L7: It would be great if the authors clarify increasing temperatures change competition (temperature-dependent competition) but do not change predation. Also please explain how increasing temperatures change competition.

L13: negative → positive

L17: Please add some concluding sentences.

P2

L37: How did they simplify species interactions?

L44: species interaction → interspecific competition

P3

L56-57: How does this change future management guidelines? I think "more trait variation is better for persistence" is rather traditional way of thinking in conservation biology.

L66: It may be worth mentioning that the assumption allows species with very low abundance to evolve rapidly, which may be unrealistic.

L75: It is not clear how predators compete. Is there interference competition in addition to resource competition (as Amarasekare 2002 Proc. R. Soc. B)?

L79: How does the randomness of trophic interactions affect dynamics? What will happen with nested or module interactions?

P4

L83-84: How does the number of patches affect the results? Why did the authors consider no area difference (polar patches should be much smaller than equator patches)?

P5

L110: It may be interesting to calculate alpha, beta, and gamma diversity as well as RAC (Rank Abundance Curve).

P7

Figure 3: How did dynamics look like before the environmental change (when $t < 1000$)? Were there stationary states or transient dynamics?

P8

L144: Figure 4 → Figure 4b, 4d

L153-154: I could not understand why this occurs immediately. Please consider rephrasing it.

P10

L179: What is the "trait-driver theory"? Please add a brief explanation or move this to Discussion with detailed explanations.

P11

L182-184: Communities with temperature-dependent competition can reduce the number of extinction events (Fig. 3, 5), but at the same time they cannot track the changing optimum (Fig. 6). Why is there the (seeming) contradiction? Tracking the optima (with smaller lag) is not necessary for persistence?

L184-185: I could not understand what this means immediately. Please consider rephrasing it.

The first line of the legend of Figure 6: Please explain that larger values in the Y-axis means the low

ability of communities to track local climate conditions.

The sixth line of the legend of Figure 6: I think the p value of simulations is not meaningful (White et al. 2014 *Oikos*).

P12

L210-211: In my understanding, Schreiber et al. (2011) *Ecology* ([18]) employed Lande's approach (Lande 1976 *Evolution*) and this is different from the approach here.

L223: Is there disruptive selection in the simulations?

P13

L224: In addition to disruptive selection, disassortative mating may be needed for sexual populations to complete speciation (e.g., Doebeli et al. 2007 *Proc. R. Soc. B*).

L237: Please clarify how trait distance is related to temperature.

P14

L272: Please add citations here.

Reviewer #3 (Remarks to the Author):

GENERAL COMMENT

In this paper authors introduce a latitudinal chain landscape quantitative genetics model to study the interplay between ecology and evolution for determining species distributions and other biodiversity metrics under (partially) empirically parametrized warming. The way authors deal with the topic is at the frontier of the connection between biodiversity, species interactions and climate change. This is mostly because the connection between climate change, trait evolution, species interactions and dispersal for biodiversity maintenance remain full of uncertainties. Authors here build a framework to start disentangling (and to start understanding better the uncertainties) the biotic and abiotic factors driving trait evolution and biodiversity maintenance along a latitudinal chain. The whole framework can open novel ways of developing a theory of eco-evolutionary biodiversity and food webs dynamics in changing environments. Below I outline the novelty statement, main results and the major drawbacks in the present version of the ms.

NOVELTY STATEMENT

The way authors combine effects is quite novel: Quantitative genetics recursion models of discrete locations (which has been classic in pop gen --- Rousset, F. (2004). *Genetic Structure and Selection in Subdivided Populations*. Princeton Univ. Press.) connected to species interactions within and between trophic levels. Furthermore, species interactions are function of warming and so local community properties can be analyzed under one-dimensional abiotic in slow-fast trait evolution and dispersal dynamics. This mix, argue authors, keep the model at the important biological details while makes it easy to analyze, interpret and it is also computationally efficient. These are important properties to build reproducible and testable frameworks. Yet, the present draft is not sufficiently clear in a couple of themes determining the generality of the framework: the complexity-tractability trade-offs and the weak selection limit approach. First, authors mention naive solution schemes still prevail in the literature, specially some concerning IBM or related methods that despite being able to include arbitrary levels of complexity tend to be computationally expensive. However, recent (fast) methods dealing with

large complexity have emerged in explainable-AI and deep learning networks - see for example, Hutson, M. (2020). AI shortcuts speed up simulations by billions of times, *Science*, 367:728; Rackauckas C. et al (2020). Universal Differential Equations for Scientific Machine Learning, arxiv.org/abs/2001.04385 - can authors build on how such novel methods might help to complement-enrich the one developed here to foster biodiversity-climate change research? Second, the whole approach is based in the weak selection limit assumption, a nearly-neutral approach with a lot of power to derive analytical expressions and to speed up simulations. Authors use it here mostly as an argument to speed up simus. There are no analytical derivations from which empirical patterns could be compared (i.e., ABC, Bayesian framework or others) against the weak selection limit scenario. Furthermore, authors mention the effect of selection on phenotypes in different parts of the main but the connection between the weak selection limit and the selection mode and strength they are dealing with in their simus is never made. How do the introduced results change under different selection strengths and selection modes? Selection is almost completely ignored in the whole result and discussion section (weak selection limit is mentioned four times in the SI and never mentioned in the main text).

MAIN RESULTS

Previous studies have found increasing extinction rates when combining trait evolution and dispersal or increasing dispersal dynamics. Here authors found that top down interactions and temperature-dependent competition alter species responses to warming resulting in species coexisting to higher degree and larger range breadths (Figures 3 and 4), with lower levels of species turnover and global losses being less severe than previous connections between trait evolution and dispersal. Authors also show community wide dispersion of species temperature optima is a strong predictor of a community capacity to respond to climate change and that this property is independent to available genetic variance and dispersal ability (Figure 6).

SIMUS

Authors run a (quasi) fully factorial combination of fast and slow average dispersal rates (see below question about how fast and slow connects to large and small), fast and slow trait evolution and the 4 following ecological models to compute species range breadths, global and local species diversity, changes in community composition induced by climate change and interspecific community wide trait lag.

1. Metacommunity constant, patch- and temperature-independent competition between species
2. Two trophic levels
3. Metacommunity temperature-dependent competition (based on species temperature optima)
4. Two trophic levels as well as temperature-dependent competition

Initial conditions were randomly drawn from prespecified distributions and authors present results from 16 combinations and 10 replicates from each one. Authors should more clearly state why only 10 replicates, were they sufficient enough to obtain robust metrics? How was robustness analyzed?

MAJOR DRAWBACKS

0. Authors argued they keep the model at the important biological details while makes it easy to

interpret and computationally efficient. They compare this "model trait" with the lack of this feature for IBM. However, authors only explore the weak selection regime numerically without any analytical derivation for the presented results, as most IBM models are analyzed. Can authors contrast predictions from the weak or neutral selection regime with explicit selection strength and selection mode? Which are the main expectations under weak selection? Do authors still think their analysis differ from an IBM analysis? Why? (this is a more specific comment following the Novelty statement above)

1. Empirically supported species temperature optima. How strong it is? Examples?

2. Climate is symmetric around the equator -- how does it relate to symmetric dispersal? and what is the connection between fast-slow to large-small dispersal? Why not using a distance dispersal kernel? Please clarify how symmetric migration is, specifically, how m_{ij} differs from m_{ji}

3. Magic trait evolution?

How do the one dimensional trait responding to warming relate to the trait responding to competition or predation? Can authors discuss this assumption a bit more around magic trait? This relates to the biotic and abiotic frequency dependent causes within and between species that might be driving some of the patterns underlying model assumptions but never are clearly explained. I am not suggesting authors implement a method to disentangle relative effects, but the reader now can miss the relative importance of the strength of selection and selection mode for the biotic or abiotic factors determining the expected species-phenotype abundances per site for the different scenarios.

MINOR COMMENTS

Figure 1 : very long caption with many details that might add confusion instead of clarity to the reader. For example, why should high growth rate individuals within the green species leave to adjacent patches? And why should low growth rate individuals immigrate? One might think emigration of common best fit phenotype is governed by larger intraspecific competition? and viceversa for the rare phenotypes? This is a complex topic to explain in a caption because frequency-dependent biotic and abiotic selection within and between species are at play and there is yet not background in the paper to go so deep at this caption. I would simplify it adding only what is new in this model and moving all the rest to the modeling framework section.

Parameter names main and SI

annual dispersal is never mentioned in the main but two times in the SI why? Please check

Ref 5 last author name

Response to reviewer comments

Reviewer 1

The authors study how different models of ecological communities and varying genetic diversity and dispersal ability influence community response to sustained environmental change.

Overall I think the authors have chosen an exciting line of research and questions. I think there are still many exciting and important lessons to be learned about ecological and evolutionary processes that underly how biodiversity at multiple levels will respond to environmental change. The authors have made important advances over previous work by increasing the realism or complexity of their evolutionary community models. However, I am concerned that despite this advance, the paper lacks a coherent central message with key lessons learned. Rather, I find that paper feels a bit sprawling. Admittedly it is challenging to find a pithy lesson in the tangled bank of complex community eco-evo responses to environmental change. Nevertheless, I think the exercise of communicating these findings would be greatly improved by developing a clear, well-connected thread of lessons learned.

We thank the reviewer for the thorough engagement with our work, the encouraging words, and the constructive criticisms. The feeling that the paper was sprawling was shared by Reviewer 2 as well. Building on the Reviewer's suggestion below, we now organize the manuscript around the idea of adding interaction complexity to models of eco-evolutionary dynamics under climate change. This takes two forms: trophic interactions and temperature-dependent competition. We thread our message around the effects these features have on community patterns in response to climate change, in comparison with earlier work that has studied the same kinds of questions but without the same interaction complexity.

The Introduction doesn't set the stage with a strong conceptual framework, it reads more like a laundry list (without strong organization) of potentially important processes and modeling issues to consider. "include how species interactions themselves change because of increasing temperatures" (L44) – this idea may be what the authors want to build their organization around. This issue comes up as well with the response variables of interest – is there a way to succinctly group and organize these?

We have now strengthened the thread of our message by focusing on the two sources of interaction complexity in our approach: trophic interactions, and how species interactions themselves change because of increasing temperatures.

This manuscript does not benefit from a Results section before the Methods. It is hard to interpret the differences in the different models without some more in depth description of those models, that may naturally lead to hypotheses.

This is very true. The difficulty is in keeping the Modeling framework section (that comes before the Results) sufficiently informative, while still relegating technical details to the Methods that are likely only of interest to a smaller readership. To address this problem, we have expanded our explanation of the various models, giving a better feel for what one can expect from them (lines 50-58 and 124-129). We also illustrate the typical dynamical outcome of each model, to further enhance the reader's intuition about what the dynamics under these various model assumptions look like—see our new Figure 3 and comment just below.

I also have trouble with the way in which results are presented. They are generally abstracted to a high level and then described in very broad terms, and collapsing down a lot of parameter variation down to a few

categories of process and pattern. There are benefits to this approach, to help avoid getting lost in too much detail, but I find that without some examples of in depth discussion of specific results, it is harder for the reader to firmly grasp what underlies the patterns identified here.

We agree that our reported results are very high-level and aggregated, making it difficult to see where they actually come from. To help with this, we now include figures of the actual community dynamics along the latitudinal gradient, to show how different model setups lead to different outcomes (Figure 3 in the main text). Our aggregated, high-level results are averages over replicates of many similar outcomes. The new figure serves the dual purpose of building intuition about what the community dynamics look like, and of motivating the questions we ask about them (lines 158-168).

I disagree with this interpretation of trait lag patterns, where the authors seem to suggest there is no predictable effect of genetic variance or dispersal on trait lag: “Interestingly, the same clustering of results is absent with respect to model parameterization in terms of available genetic variance and dispersal ability.” (L185). It is clear from Fig 6 that there are effects of genetic variance and dispersal. For example, nearly all the '+' fall above the trend line, indicating greater lag than expected when there is low genetic variance and low dispersal ability, regardless of the type of community model. Circle symbols fall nearly all below the trend line, indicating less lag than expected for high genetic variance and high dispersal.

The Reviewer is completely right, and our text was unclear on this point to begin with. By “clustering”, we mean that points of a given color (corresponding to model type) tend to fall around more or less well-defined trait dispersion values. For example, the baseline model (blue; Figure 7) is clustered at low average trait dispersion values. In addition to this, model parameterization influences whether the points fall above or below the regression line. We have changed our wording to make this clear (lines 221-234). Please also note that, since we have increased simulation times before the start of climate change from 1000 to 4000 years, this has altered some of the quantitative aspects of the figure, though the basic negative trend is retained. In particular, it is now clear that changing the parameter values does in fact influence the slope and intercept of the regression line.

I am confused a bit by how with temperature dependent competition species “coexist to a higher degree” (L231) yet there is also lower trait diversity within communities (Fig 6). How does more coexistence and species diversity also lead to lower trait diversity?

This seemingly paradoxical result stems from a comparison of local and global patterns. With temperature-dependent competition, more species coexist locally in any given patch. However, their trait values across the patches are less variable—precisely because they do not need to adapt as closely to the local temperatures as in the temperature-independent case. Whereas, in that latter case, even though there are fewer species per patch (and thus global diversity may also be lower), their temperature optima are more variable across the landscape, leading to higher trait dispersion overall.

The last two sentences of the Abstract should be worded more clearly. They're a bit confusing. We have reworded the sentences (lines 14-19).

L45 – I'd reword to say 'has been little considered' or something like that, as opposed to the strong priority claim.

We changed the sentence (lines 50-51).

L141 – the point here is unclear – from the wording, I cannot tell whether there is a scenario where cold adapted species expand ranges or whether there is no such scenario.

There is no scenario in which the most cold-adapted species expand their ranges. This expansion only happens to warm-adapted species. While this was indicated in the following sentence of our text, we have now restructured them to avoid any confusion (lines 187-191).

L179—perhaps worth noting that this hypothesis is directly analogous to Fisher’s fundamental theorem

The connection with Fisher’s fundamental theorem is indeed attractive—we thank the Reviewer for pointing this out. In our case, an even better analogy is provided by the secondary theorem of natural selection (Robertson 1968): the response of some trait z of a species is proportional to the covariance of the trait in the population and the fitness values r over it: $dz/dt \sim \text{cov}(z, r)$. Fisher’s fundamental theorem is a special case, where the trait we consider is itself the fitness: $dr/dt \sim \text{cov}(r, r) = \text{var}(r)$. In our case, the evolving trait of interest is the temperature optimum, while each optimum value has some fitness determined by the environment and species interactions. We now mention in our text that the secondary theorem provides a simple argument for expecting the observed outcome (lines 215-218).

Figure 1 would benefit from some more labeling on the figure itself – especially the right panel. Labels on the curves shown could help. It is also a bit confusing that the species shown here are maladapted to the local temperature.

We have added labels to the figure, and placed the focal species at the temperature optimum of the local patch. This also makes it more credible why the immigrants’ traits are slightly off: since neighboring patches have slightly different temperatures, it stands to reason that the focal species is adapted to temperatures in the neighboring patches, meaning their immigrants are slightly maladapted when compared with the focal temperature.

Figure 4 would benefit from a label on the x-axis or in the margin that indicates which species are polar and which tropical.

We have now labeled the x-axis in the figure (which became Figure 5 in the revision), going from polar to tropical species.

Reviewer 2

Recent studies have demonstrated the importance of rapid evolution, species interactions, and dispersal for understanding the effects of climate change on biodiversity. This manuscript constructed a simulation model where discrete patches are connected by dispersal, and species evolve in response to climate change with competition and predation. The authors found that predation and temperature-dependent competition could mitigate the negative effect of climate change. Furthermore, large trait variance within the community could reduce the lag between the community mean trait and environmental optimum. Because of the importance of predicting and understanding eco-evolutionary dynamics under climate change, this study is of interest to many people in this research area. However, I feel the manuscript has some weaknesses to be addressed.

We thank the Reviewer for the thorough reading of our manuscript and the helpful comments. We especially appreciated the suggestion of exploring consumer generality (which we now do), and the question about the pre-climate change period. This forced us to look at the pre-climate change dynamics more closely, and we decided to increase its duration from the original 1000 to 4000 years to achieve better convergence (see below for details).

Major comments:

1. *What are surprising results of this study? It is well known that specialist predators can promote prey coexistence (as kill-the-winner hypothesis), and larger trait variation allows rapid responses to changing environments (e.g., Kopp & Matuszewski 2014 Evol. Appl.). And the mitigating effect of predation and temperature-dependent competition on extinction may be examined without considering dispersal. If this paper is to construct a new modeling framework (by combining quantitative genetic modeling and discrete spatial structure instead of using PDEs with continuous space), it may be better to publish in, for example, Methods in Ecology & Evolution.*

Clearly, our manuscript is at the boundary of a methodological and a research contribution. It is methodological in that the way we incorporated trophic interactions and temperature-dependent competition is an important part of our message; however, we also look at the consequences of these mechanisms, contrasting them with more or less known outcomes when such complications are absent (e.g., Norberg et al. 2012, Thompson and Fronhofer 2019, Lasky 2019). We would say that this is where the novelty of our study lies: how do trophic interactions and temperature-dependent competition influence eco-evolutionary dynamics under climate change? How do they act when they operate alone, and how do they interact when both are present simultaneously?

It is indeed known that specialist predators can promote prey coexistence. By itself, this would not be surprising. In our model, each predator is connected to five prey species (the one with similar initial temperature optima, and four randomly assigned). An interesting question is whether and how trophic and spatial dynamics interact with one another, leading to outcomes that are different from what one would expect purely on the basis of predator-mediated coexistence (or exclusion via apparent competition). To go deeper into exploring this, we have re-run our analyses with a dispersal rate of zero: this way, we have a series of isolated patches. The results are included in our revised Supplementary Information (SI; Section 5.4). In summary, the mitigating effect of, in particular, temperature-dependent competition still appears even without dispersal. Dispersal does however play an important role, which becomes evident when we compare our zero-dispersal setup with the original one, including two levels of dispersal. The possibility to disperse, in combination with fast evolution, clearly exaggerates global extinctions compared to when species cannot disperse. Also, no dispersal results in narrower range breadths and lower local species diversity than in the presence of dispersal. Thus, we find that the interplay between ecological (dispersal and species interactions) and evolutionary (adaptation to new conditions) processes along a spatial gradient do significantly affect species response to altered climatic conditions in less-known and unforeseen ways. Additionally, we have re-run our analyses, controlling for predator generality. The results are included in our revised SI (Section 5.3) and are explained more in detail under Reviewer comment no. 5 below.

2. *It is not clear how temperature affect competition in Abstract and Introduction. Based on Equation 6 and the explanations in L321-323, readers who are familiar with math can see how it works, but they may still wonder why the competition coefficient, a , is not a function of temperature, T . Please add careful explanations to Abstract and Introduction as this is one of the most important point of this study. Although it seems that the authors assumed that temperature-dependent competition reduces the strength of interspecific competition (L153-154), it may be possible to see stronger interspecific competition due to temperature-dependence, right?*

This is a good point. While the Abstract does not leave much room for a detailed explanation, we have modified the Introduction to explain this point better (lines 50-58).

In general, it is of course possible that competition between two species is an increasing (as opposed to decreasing) function of the difference between their phenotypes. For example, allelopathic plants produce

toxins to poison their competitors, but they are immune to their own toxins. In that case, the strength of competition conceivably increases as a function of toxin quality difference, since two species producing very similar toxins are mutually (quasi-)immune to one another, whereas species with very different toxins are not. It is more difficult though to think of biologically realistic scenarios where competition would increase as a function of the difference in temperature tolerance. The reason is that temperature tolerance primarily governs the type of habitat available for the species, so any difference in temperature tolerance leads to a partitioning of suitable habitat in heterogeneous environments, lowering competition. It is of course true that when two individuals compete for the same site, then one of them will win, and if one species is much better adapted to the site, it will win more easily. This, however, is not a consequence of heightened competition, but the fact that the maladapted organism cannot grow at that location in the first place. (In terms of a simple Lotka–Volterra competition model: the competition coefficients need not differ, but the intrinsic rates and carrying capacities do.)

3. As shown in Figure 1, dispersal from neighboring patches can result in multimodal distribution of species traits, but the authors explained that they assumed unimodal, normal trait distributions (L221-226). How did the authors use the quantitative genetics framework to deal with this? Please add explanations to Introduction.

We have now done so (in the Modeling framework section; lines 81-89). The key idea is that the momentary multimodality arising from immigration disappears after one round of random mating, which regenerates the normal shape of the distribution. This works in the same way as restoring normality after selection: even if selection removes, say, all individuals below some threshold temperature optimum (thus resulting in a starkly nonnormal trait distribution), the assumptions of random mating and many additive loci with small individual effects guarantee that the distribution will again be normal after just a single generation (e.g., Bulmer 1980, Barton et al. 2017).

4. Figures 3, 4, and 5 have 16 small figures and it is sometimes unclear which figures the authors are talking about. It will be great if the authors can add characters (a, b, c,...) to the small figures so that readers can understand which is which.

We have now labeled all subfigures, and use these labels to refer to particular subfigures as necessary throughout the text.

5. Although the authors concluded that predation can reduce extinction events, it is not clear how the generality of predators affects this outcome. While specialist predators promote negative density-dependence in community dynamics and cause coexistence, generalist predators may promote apparent competition and make coexistence difficult. It will be great if the authors can change the generality of predators to see the robustness of their results.

This is an excellent point. We have generated further model runs, controlling for predator generality. The results are included in our revised SI (Section 5.3). These are consistent with the idea that specialist consumers promote higher species diversity with a higher level of coexistence. However, enhanced coexistence is retained in the generalist case as well, and local losses are less severe with generalists than without any consumer presence. This is especially visible when consumer presence is combined with temperature-dependent competition. In conclusion, while specialist consumers do promote diversity more than generalist ones, even generalists promote it to an extent.

0. The authors considered competition and predation as species interactions, but how do mutualistic interactions change eco-evolutionary dynamics under climate change? Northfield & Ives (2013) PLOS Biol.

showed that coevolution with mutualism can destabilize dynamics whereas coevolution with competition and predation tend to stabilize dynamics. It will be great to discuss the perspectives in Discussion.

This is an important problem, and while addressing it in depth is beyond the scope of our study, we now mention the possible effects of mutualistic interactions in the Discussion (lines 290-299).

7. It would be great if the authors focus on a single patch and examine whether the same patterns reported in previous studies on evolutionary rescue with interspecific interactions occur or not. For example, competition (de Mazancourt et al. 2008 Ecol. Lett., Osmond & de Mazancourt 2013 Phil. Trans. R. Soc. B) and predation (Osmond et al. 2017 Am. Nat., Cortez & Yamamichi 2019 Ecology) may promote or prevent extinction depending on model assumptions and parameters.

As previously mentioned, we have re-run our analyses with a dispersal rate of zero: this way, we have a series of isolated patches, within which the species that are locally present can still interact. We have also considered these results in light of the literature suggested by the Reviewer above. These are now included in our revised SI (Section 5.4).

Osmond and de Mazancourt (2013) show that interspecific competition can aid evolutionary rescue when it speeds up evolution by increasing the strength of selection enough to overcome the negative effect of reduced abundances. In this study, the focal species evolve towards the temperature optima, the pace being dependent on the competition. We have no model setup without competition, since species compete in all our models (even the baseline one). For this reason, direct comparison of our results with those of Osmond and de Mazancourt (2013) is not possible. In our study, temperature-dependent competition implies that most species do not evolve to match the temperature optimum of a patch (they instead find a suitable niche more or less off the temperature optimum). We show that temperature-dependent competition facilitates adaptation and increases persistence to a higher extent than temperature-independent competition (Figure S17).

De Mazancourt et al. (2008) compare A) single species in a single patch, B) single species in various numbers of patches, C) Multiple species in multiple patches (that compete and disperse). Scenario C resembles our baseline scenario with dispersal. De Mazancourt et al. (2008) show that with multiple species, they are increasingly restricted to only those patches with temperature optima close to their initial phenotype, since occupation of more dissimilar patches are prevented by the presence of other species. With many species present, the amount of evolutionary change decreases, and extinction rates increase. Our baseline setup similarly shows that few species coexist locally (Figure 4), and global losses are severe (Figure 6).

Similar to our results, Osmond et al. (2017) find that consumer presence reduces resource densities (Figure 3). With consumer presence but without dispersal, we observe slightly boosted local resource richness and increased ranges, in comparison to what we see without consumers (Figure S15-S16). Contrasting with Osmond et al. (2017), we observe a similar but marginal effect of consumer presence, with slightly fewer global losses when evolution is slow (Figure S17).

Cortez and Yamamichi (2019) focus on how evolution in prey, predators, or both, affect responses of predator populations to an increase in mortality. Both synergistic and antagonistic effects between prey and predator evolution appear, but compared to their scenario with no evolution at all, evolution of one or both species increases the predator mortality extinction threshold in most cases. What we observe is that higher levels of available genetic variance (leading to faster evolution) actually slightly decreases consumer persistence (Figure S18). While this appears to contrast with the observed increase in mortality thresholds by Cortez and Yamamichi (2019), they in fact have shown that consumer density

can temporarily *increase* as a function of increasing mortality (hydra effect). For this reason, the outcome of increasing the speed of evolution is not immediately obvious in our multispecies setting. The fact that we find a slight reduction in consumer persistence is therefore not necessarily inconsistent with their findings.

Minor comments:

P1

Title: It is hard to know the results of this study from the current title. What about “Specialist predation and temperature-dependent competition mitigate the negative impact of climate change on biodiversity” or something like this?

We certainly like the suggested title. However, it might put the focus on only one part of the results, because it does not mention the eco-evolutionary perspective. A spatially explicit eco-evolutionary model with 2 different types of species interactions (trophic and competitive) and a trait-based perspective on top of this is bound to generate many insights and we feel that the suggested title might not capture their full range. With the Reviewer’s permission, we would thus prefer to keep the original title, and rely on waking the reader’s interest enough with it to read through the abstract which lays out the main findings.

Authors: What is the affiliation of Bo Ebenman?

Bo Ebenman has recently retired. We have added his affiliation that was valid right before retirement.

L5: What is the difference between species- and community-level dynamics? Did the authors mean population-and community-level dynamics?

Yes, that is correct; however we decide to re-write that paragraph and removed that sentence.

L7: It would be great if the authors clarify increasing temperatures change competition (temperature-dependent competition) but do not change predation. Also please explain how increasing temperatures change competition.

We have now done so, in the Modeling framework section (lines 126-129; see also the Introduction, lines 50-58).

L13: negative → positive

Thank you for catching that; we have changed it (line 13).

L17: Please add some concluding sentences.

We have done so (lines 17-19).

P2

L37: How did they simplify species interactions?

Norberg et al. (2012) use the simplest possible interaction structure: each species interacts with every other (including itself) at the exact same interaction strength. That is, were the intrinsic growth rates of these species also equal, they would coexist neutrally. Lasky (2019) employs diffuse competition: there is a single common intraspecific competition coefficient, and also a single interspecific coefficient common to all species interactions. We now mention this in the text (lines 40-42).

L44: species interaction → interspecific competition

Changed (lines 47-48).

P3

L56-57: How does this change future management guidelines? I think “more trait variation is better for persistence” is rather traditional way of thinking in conservation biology.

We neglected to explain this before, so we now include a new paragraph in the Discussion detailing this point (lines 345-353). Briefly: focusing on trait driver theory simplifies the identification of conservation strategies underpinning the maintenance of trait variation in some trait, and thus a particular environmental driver of concern. For example, ensuring connectivity to habitats with higher mean temperature or temperature variation can promote an influx of species (or genotypes) that can cope with an increasing trend in temperature by maintaining the local trait variation of temperature optima. This perspective therefore complements and refines the traditional conservation viewpoint that “more biodiversity is better”.

L66: It may be worth mentioning that the assumption allows species with very low abundance to evolve rapidly, which may be unrealistic.

This is an important point. To avoid the rapid evolution of species with low abundance, we apply a reduction to the heritabilities h^2_i at low population densities. This was described in the SI (Section 3.4); we now also mention it in the main text for clarity (lines 84-85).

L75: It is not clear how predators compete. Is there interference competition in addition to resource competition (as Amarasekare 2002 Proc. R. Soc. B)?

There is no direct predator interference: consumers compete only indirectly, via shared resources. This statement was buried at the beginning of the SI's Section 3.2, but now we also clarify it in the text (line 97).

L79: How does the randomness of trophic interactions affect dynamics? What will happen with nested or module interactions?

These are very important questions to which we do not know the answer yet. One could use our model to look at this, by designing a numerical experiment where network modularity and/or nestedness are gradually altered, keeping other things equal. However, such an investigation is well beyond the scope of our study. Just as with exploring the influence of mutualism, we therefore leave this problem as future work (lines 290-299).

P4

L83-84: How does the number of patches affect the results? Why did the authors consider no area difference (polar patches should be much smaller than equator patches)?

Beyond a certain patch density, results no longer change by adding even more patches. In our case, having 50 patches was sufficient to achieve this convergence in the results. We show this in a new subsection in the SI (Section 5.7), where we include figures of local diversity, species ranges, and global species richness that were run on 100 instead of 50 patches. The results are just as those obtained with only 50 patches (Figures 4-6 in the main text).

The fact that polar patches are smaller than equatorial ones is, in a sense, already incorporated in the model. Figure 2A shows that more polar species have lower intrinsic growth rates. In our model, the carrying capacity (monoculture equilibrium density) of a species is directly proportional to the intrinsic rate: taking Eq. S36 without any other species or migration, it reduces to $dN/dt = N(b - \alpha N)$, with the nontrivial equilibrium reading $N^* = b/\alpha$. Since b is related to the integral of the curves in Figure 2A (Eq. S32), this results in polar species having lower carrying capacities—which in turn can be interpreted as polar patches being smaller.

P5

L110: It may be interesting to calculate alpha, beta, and gamma diversity as well as RAC (Rank Abundance Curve).

While we did not call them by those names (we now do; line 138), our Figures 4 and 6 already accounted for alpha (local) and gamma (global) diversity. That said, adding an analysis of beta diversity and rank-abundance curves is a valuable idea, which we have now done (SI, Section 5.2). The beta-diversity pattern (Figure S7) exhibits a peak shortly after climate change stops, for every scenario. It also shows that there is less species turnover when we have temperature-dependent competition and/or trophic interactions included, except when both evolution and dispersal are slow. The rank-abundance curves at the end of the simulations follow a characteristic S-shape known from the literature (Figure S8).

P7

Figure 3: How did dynamics look like before the environmental change (when $t < 1000$)? Were there stationary states or transient dynamics?

We thank the Reviewer for this question, which made us double-check the pre-climate change results. What we have found is that 1000 years were often insufficient to get close to a steady state. This may, of course, have confounding effects on our results, which then could possibly be artifacts of transient effects.

To control for this, we have increased the pre-climate change period from 1000 to 4000 years. This was sufficient to bring the system either to, or very close to, a steady state (see Figures R1-R2 below). We have correspondingly re-simulated and revised all our figures and results which an increased, 4000-year-long establishment phase. Compared to what we had before, this resulted in only one qualitative change to the results: the addition of trophic interactions has only a small positive effect global species richness relative to the same model setup without the second trophic level (a strong positive effect on local diversity is retained on the other hand). Apparently, the positive effect was due to transients and not trophic interactions *per se*. We have now corrected for this in our text as well, wherever it was appropriate.

As seen in Figure R2 below, in some cases even 4000 years were insufficient to completely reach the steady state. Therefore, to triple-check our results, we also ran some simulations with not just a 4000, but a 20000-year establishment period. While this did result in reaching (or almost reaching) the actual steady state, it made no difference to the actual post-climate change results whatsoever. For this reason, we kept with the computationally more efficient 4000-year establishment period.

As the duration of the establishment phase was now increased, we also adopted a different convention for measuring time. Now, $t_0 = -4000$ years marks the start of the simulations, $t = 0$ is the start of climate change, $t_E = 300$ years is when climate change ends, and simulations last until $t = 2500$ years. So the simulations now encompass a 6500-year period altogether, instead of the original 3500 years.

P8

Figure R1: As Figure 4 in the main text, but with species richness before the start of climate change ($-4000 < t < 0$).

L144: Figure 4 → Figure 4b, 4d

Correct; we have added the labels and now use them in referring to the figures wherever necessary.

L153-154: *I could not understand why this occurs immediately. Please consider rephrasing it.*

We have rephrased the sentence (lines 173-178). The point is that, with temperature-dependent competition, the advantages of being sufficiently different from other locally present species can be so great that they outweigh the disadvantages of being somewhat maladapted to the local temperatures.

P10

L179: *What is the “trait-driver theory”? Please add a brief explanation or move this to Discussion with detailed explanations.*

Figure R2: As Figure 6 in the main text, but with species richness before the start of climate change ($-4000 < t < 0$).

We have added an explanation and reference (lines 219-220).

P11

L182-184: *Communities with temperature-dependent competition can reduce the number of extinction events (Fig. 3, 5), but at the same time they cannot track the changing optimum (Fig. 6). Why is there the (seeming) contradiction? Tracking the optima (with smaller lag) is not necessary for persistence?*

When competition is temperature-dependent, species need not track the environmental optimum, because being sufficiently different from a dominant species already occupying the optimum will result in the reduction of competition, and thus the species can survive even though they are maladapted to local temperatures.

L184-185: *I could not understand what this means immediately. Please consider rephrasing it.*

Our text was indeed unclear on this point, as was also pointed out by Reviewer 1. By “clustering”, we meant that points of a given color (corresponding to model type) tend to fall around more or less well-defined points along the trait dispersion axis. Model parameterization, on the other hand, influences whether the points fall above or below the regression line, modifying its slope and intercept. We have changed our text to make this clear (lines 221-234).

The first line of the legend of Figure 6: Please explain that larger values in the Y-axis means the low ability of communities to track local climate conditions. The sixth line of the legend of Figure 6: I think the p value of simulations is not meaningful (White et al. 2014 Oikos).

We have added the explanation to the legend. And yes, we have removed the reported *p*-value.

P12

L210-211: In my understanding, Schreiber et al. (2011) Ecology ([18]) employed Lande’s approach (Lande 1976 Evolution) and this is different from the approach here.

Broadly speaking, both Lande (1976) and Schreiber et al. (2011) belong in the same class of models that rely on quantitative genetics to model the evolutionary consequences of selection. Our model is also part of this same family. Ultimately, however, ours is quite distant from its origins in Lande (1976): it is in continuous time, with explicit frequency-dependent selection, multiple species, and multiple patches connected by dispersal.

L223: Is there disruptive selection in the simulations?

There is, but only with temperature-dependent competition. Take a single species in a given patch. It will evolve towards the environmental optimum. However, being different from the dominant phenotype may be so advantageous as to make the optimal strategy pessimal by the time it is reached (e.g., Dieckmann and Doebeli 1999, Meszena 2005). This results in disruptive selection at that (former) optimum point. In models of adaptive dynamics, the species could then split into two daughter species which evolve away from one another. In our case, since trait distributions are forced to remain normal, such a split does not happen.

P13

L224: In addition to disruptive selection, disassortative mating may be needed for sexual populations to complete speciation (e.g., Doebeli et al. 2007 Proc. R. Soc. B).

Good point; we have modified the sentence to mention this (lines 269-271).

P14

L272: Please add citations here.

We have added the references and clarified our text (lines 325-336).

Reviewer 3

GENERAL COMMENT

In this paper authors introduce a latitudinal chain landscape quantitative genetics model to study the interplay between ecology and evolution for determining species distributions and other biodiversity metrics

under (partially) empirically parametrized warming. The way authors deal with the topic is at the frontier of the connection between biodiversity, species interactions and climate change. This is mostly because the connection between climate change, trait evolution, species interactions and dispersal for biodiversity maintenance remain full of uncertainties. Authors here build a framework to start disentangling (and to start understanding better the uncertainties) the biotic and abiotic factors driving trait evolution and biodiversity maintenance along a latitudinal chain. The whole framework can open novel ways of developing a theory of eco-evolutionary biodiversity and food webs dynamics in changing environments. Below I outline the novelty statement, main results and the major drawbacks in the present version of the ms.

We thank the Reviewer for the encouraging comments, helpful suggestions, and interesting ideas to improve our manuscript. Please see our detailed responses below.

NOVELTY STATEMENT

The way authors combine effects is quite novel: Quantitative genetics recursion models of discrete locations (which has been classic in pop gen — Rousset, F. (2004). Genetic Structure and Selection in Subdivided Populations. Princeton Univ. Press.) connected to species interactions within and between trophic levels. Furthermore, species interactions are function of warming and so local community properties can be analyzed under one-dimensional abiotic in slow-fast trait evolution and dispersal dynamics. This mix, argue authors, keep the model at the important biological details while makes it easy to analyze, interpret and it is also computationally efficient. These are important properties to build reproducible and testable frameworks.

We thank the Reviewer for this positive overall evaluation of our approach.

Yet, the present draft is not sufficiently clear in a couple of themes determining the generality of the framework: the complexity-tractability trade-offs and the weak selection limit approach. First, authors mention naive solution schemes still prevail in the literature, specially some concerning IBM or related methods that despite being able to include arbitrary levels of complexity tend to be computationally expensive.

We emphasize that we did not and do not criticize individual-based methods (IBM) on the grounds of naive solution schemes. We confined that criticism strictly to solution schemes for partial differential equations (Press et al. 2007, ch. 20). Individual-based schemes are usually perfectly fine in terms of numerical stability—but, of course, they are computationally quite intensive, as one must keep track of every individual separately.

However, recent (fast) methods dealing with large complexity have emerged in explainable-AI and deep learning networks – see for example, Hutson, M. (2020). AI shortcuts speed up simulations by billions of times, Science, 367:728; Rackauckas C. et al (2020). Universal Differential Equations for Scientific Machine Learning, arxiv.org/abs/2001.04385 – can authors build on how such novel methods might help to complement-enrich the one developed here to foster biodiversity-climate change research?

Emerging methods such as universal differential equations may indeed be game changers in approaching numerical problems—in ecology as well as meteorology, climate research, and other areas of science. That said, there are good reasons for caution at this point in time. First, this is not a published method. The preprint of Rackauckas et al. (2020) has not been through peer-review yet, and while this is likely only a matter of time, important changes may still be made to the material in the paper. Second, in basic science, algorithm reliability is an important virtue. By this we mean the use of algorithms that are built on mathematical considerations which, when used correctly, guarantee that the method returns the right results. Notably, methods based on machine learning often do not fulfill this criterion. Neural network models for facial recognition are known to make mistakes. And even the revolutionary *AlphaGo* computer

program, which beat 9-dan Go player Lee Sedol 4 to 1 in a 5-game match in 2016, lost its one game due to a known quirk in the neural network which caused it to unduly overestimate its position under some circumstances (Sadler and Regan 2019). While universal differential equations and scientific machine learning most likely represent the future, at this point (and in relation to our manuscript) its consideration would be far too premature. That said, it is indeed a very promising direction, which we now mention in the discussion (lines 259-261).

We would also like to emphasize that the possibility of universal differential equation methods does invalidate or trivialize the computational advantages of our approach. Even if such methods have the potential to speed up simulations, such a speed-up is presumably relative to the original speed. So a method that is a hundred times faster than another will retain its relative advantage. Thus, the faster method will still allow one to perform a hundred times more simulations in a given amount of time, regardless of the fact that both methods are now faster in an absolute sense.

Finally, the preprint of Rackauckas et al. (2020) strictly concerns itself with differential equations of various kinds. While the same method might be extensible to individual-based approaches, it is not immediately clear if this is possible, and if so, how much the original method would need to be updated. If one keeps an eye on potentially applying the universal differential equation approach, then this is an argument for sticking with differential equation models.

Second, the whole approach is based in the weak selection limit assumption, a nearly-neutral approach with a lot of power to derive analytical expressions and to speed up simulations.

To clarify, the weak selection limit does not imply near-neutrality. It also does not imply that trait values can only change slightly in response to selection. All it means is that selection is not so strong as to break the assumption of smoothness, so it can be represented via differential equations instead of discrete difference equations. The distinction is well illustrated by the logistic equation in discrete and continuous time: the continuous logistic's behavior is always smooth, while the discrete logistic may produce cycles and chaos for too large growth rates. Selection is "weak" (in this case, the population growth rate is sufficiently small) as long as one can accurately represent the discrete logistic's behavior using the continuous-time model instead.

Authors use it here mostly as an argument to speed up simus. There are no analytical derivations from which empirical patterns could be compared (i.e., ABC, Bayesian framework or others) against the weak selection limit scenario. Furthermore, authors mention the effect of selection on phenotypes in different parts of the main but the connection between the weak selection limit and the selection mode and strength they are dealing with in their simus is never made. How do the introduced results change under different selection strengths and selection modes? Selection is almost completely ignored in the whole result and discussion section (weak selection limit is mentioned four times in the SI and never mentioned in the main text).

We now mention the assumption of weak selection in the Modeling framework section (lines 101-104). To reiterate, working in the weak selection limit is a technical assumption which is known not to be very restrictive (e.g., Barabás and D'Andrea 2016). In fact, it allows one to look at the effects of quite strong selection, without compromising accuracy too much. We can demonstrate this explicitly on a classical model from population genetics: the replacement of an allele a by another allele A in a haploid population under constant directional selection. The equation reads

$$p(t + \Delta t) = p(t) + s p(t) \frac{1 - p(t)}{1 + sp(t)}$$

(e.g., Otto and Day 2007, ch. 3), where $p(t)$ is the frequency of allele A at time t , Δt is the generation time, and s is the selection differential. Subtracting $p(t)$ from both sides, Taylor expanding the right hand side to linear order in s , and performing the $\Delta t \rightarrow 0$ limit, we arrive at the weak selection approximation

$$\frac{dp(t)}{dt} = sp(t)(1 - p(t))$$

Figure R3 compares the behavior of the two models, for various values of the selection differential s , ranging from weak-ish to outrageously strong selection. As seen, the model in the weak selection approximation makes nearly the same predictions as the one without, regardless of the strength of selection.

Figure R3: Allelic frequency p (ordinate) against time (abscissa), in the haploid constant directional selection model. The behavior of the original, discrete-time model (solid lines) is well-matched by that of its weak-selection approximation (dotted lines), for each selection strength (colors). This demonstrates that weak selection is more of a simplifying technical assumption that does not fundamentally alter the system’s dynamical trajectory.

The other part of the Reviewer’s comment concerns the mode of selection. This, however, is not as clear-cut as in classic population genetic models (such as the one above) which operate with highly simplified selection scenarios. In our model, species respond to local climate (frequency-independent directional selection, unless a species is at the local environmental optimum), to consumers and resources (frequency-dependent selection), and competitors (also frequency-dependent selection, possibly complicated by the temperature-dependence of the competition coefficients mediating frequency dependence). Additionally, migration to and from patches complicates all these simultaneously operating selective processes. Ultimately, there is nothing simpler to be said about the “mode” of selection in our model.

MAIN RESULTS

Previous studies have found increasing extinction rates when combining trait evolution and dispersal or increasing dispersal dynamics. Here authors found that top down interactions and temperature-dependent competition alter species responses to warming resulting in species coexisting to higher degree and larger range breadths (Figures 3 and 4), with lower levels of species turnover and global losses being less severe than previous connections between trait evolution and dispersal. Authors also show community wide dispersion of species temperature optima is a strong predictor of a community capacity to respond to climate change and that this property is independent to available genetic variance and dispersal ability (Figure 6).

SIMUS

Authors run a (quasi) fully factorial combination of fast and slow average dispersal rates (see below question about how fast and slow connects to large and small), fast and slow trait evolution and the 4 following ecological models to compute species range breadths, global and local species diversity, changes in community composition induced by climate change and interspecific community wide trait lag.

1. Metacommunity constant, patch- and temperature-independent competition between species

2. Two trophic levels

3. Metacommunity temperature-dependent competition (based on species temperature optima)

4. Two trophic levels as well as temperature-dependent competition

Initial conditions were randomly drawn from prespecified distributions and authors present results from 16 combinations and 10 replicates from each one. Authors should more clearly state why only 10 replicates, were they sufficient enough to obtain robust metrics? How was robustness analyzed?

We thank the Reviewer for the succinct summary our results and approach. We originally used 10 replicates because the summary properties we considered here (global diversity, local coexistence, and so on) exhibited almost no variation across them. In any event, we have now re-run our results with 100 replicates instead of 10, and replaced all figures. Randomly sampling 10 replicates from these 100 produces slightly noisier, but otherwise practically indistinguishable results from the full 100-replicate dataset.

MAJOR DRAWBACKS

0. Authors argued they keep the model at the important biological details while makes it easy to interpret and computationally efficient. They compare this “model trait” with the lack of this feature for IBM. However, authors only explore the weak selection regime numerically without any analytical derivation for the presented results, as most IBM models are analyzed. Can authors contrast predictions from the weak or neutral selection regime with explicit selection strength and selection mode? Which are the main expectations under weak selection? Do authors still think their analysis differ from an IBM analysis? Why? (this is a more specific comment following the Novelty statement above)

As argued above, the weak selection approximation is not restrictive, and the dynamical expectations will generally not be different when this assumption is used. (Incidentally, we never claimed that our analyses would differ from what an IBM model would predict—just that our method might get there faster.)

1. Empirically supported species temperature optima. How strong it is? Examples?

We cite Deutsch et al. (2008), basing their simulations on ectotherm data across latitudes. We added the reference Thomas et al. (2012), validating the same pattern for phytoplankton globally.

2. Climate is symmetric around the equator – how does it relate to symmetric dispersal? and what is the connection between fast-slow to large-small dispersal? Why not using a distance dispersal kernel? Please clarify how symmetric migration is, specifically, how m_{ij} differs from m_{ji}

Our dispersal matrix is symmetric: $m_{ij} = m_{ji}$ (SI, Section 3.5). Technically speaking, we are in fact using a dispersal kernel, albeit a crude rectangular one that is constant from zero to a threshold distance reaching to adjacent patches, and zero beyond. It could be replaced with other, smoother kernel types—for example, the Gaussian kernel in Figure R4. One should not forget though that our patches reach around

the globe and thus cover large distances. The rectangular kernel restricts annual movement to the vicinity of each patch, avoiding overly fast dispersal. In any event, we have now used the Gaussian kernel shown in Figure R4 to re-generate our results (with only 20 replicates instead of 100). These are shown in Figures R5-R7 below. The quantitative details are of course different, but the qualitative conclusions are the same as with the original rectangular dispersal kernel.

While we felt that burdening the manuscript with yet another set of results that do not really alter our conclusions would be overkill (so we did not include them in the manuscript), we will be happy to do so if the Reviewer feels this would be important.

Figure R4: Two different dispersal kernels, as a function of the dispersal distance x : the rectangular one used by us that is 1 for $x \leq 0.02$ and zero otherwise, and a Gaussian kernel given by $\exp(-x^2/(2\sigma^2))$, with $\sigma = 0.02\sqrt{2/\pi}$. The area under both curves is the same, 0.02. As in our text, x is measured in units of the pole-to-equator distance.

3. Magic trait evolution?

How do the one dimensional trait responding to warming relate to the trait responding to competition or predation? Can authors discuss this assumption a bit more around magic trait? This relates to the biotic and abiotic frequency dependent causes within and between species that might be driving some of the patterns underlying model assumptions but never are clearly explained. I am not suggesting authors implement a method to disentangle relative effects, but the reader now can miss the relative importance of the strength of selection and selection mode for the biotic or abiotic factors determining the expected species-phenotype abundances per site for the different scenarios.

We have added a brief discussion of this point (lines 265-273), especially that Reviewer 2 also asked for clarification regarding this. In short, our whole approach strictly assumes that the trait in question (temperature tolerance optimum) is *not* a magic trait. The fact that our trait distributions are always normal mathematically follows from the assumption of many additive loci influencing the trait and random mating. Since, by definition, a magic trait drives both species interactions and assortative mating, such a trait would violate the assumption of random mating. We now emphasize that the normality assumption rules out the possibility of temperature tolerance acting as a magic trait in our model.

As for disentangling modes of selection, see our answer above.

MINOR COMMENTS

Figure R5: As Figure 4 in the main text, but with a Gaussian dispersal kernel.

Figure 1 : very long caption with many details that might add confusion instead of clarity to the reader. For example, why should high growth rate individuals within the green species leave to adjacent patches? And why should low growth rate individuals immigrate? One might think emigration of common best fit phenotype is governed by larger intraspecific competition? and vice versa for the rare phenotypes? This is a complex topic to explain in a caption because frequency-dependent biotic and abiotic selection within and between species are at play and there is yet not background in the paper to go so deep at this caption. I would simplify it adding only what is new in this model and moving all the rest to the modeling framework section.

We have revised, shortened, and streamlined this caption (see also the changes we made in response to the comments of Reviewer 1).

Parameter names main and SI: annual dispersal is never mentioned in the main but two times in the SI why? Please check

Figure R6: As Figure 5 in the main text, but with a Gaussian dispersal kernel.

We always mentioned dispersal without the qualifier “annual”, both in the main text and the SI; the only exception was in Tables S1-S2, summarizing the meaning and units of various model quantities. We did so to emphasize the units of the dispersal rates, since dispersal could, in principle, be measured in any other units as well. To avoid confusion, we have replaced “annual dispersal distance” with “dispersal rate” in Tables S1-S2.

Ref 5 last author name

We are not sure what the Reviewer’s question is. We have checked Ref 5, and it appears correct.

Figure R7: As Figure 6 in the main text, but with a Gaussian dispersal kernel.

References

- Barabás, G., D’Andrea, R., 2016. The effect of intraspecific variation and heritability on community pattern and robustness. *Ecology Letters* 19, 977–986.
- Barton, N. H., Etheridge, A. M., Véber, A., 2017. The infinitesimal model: Definition, derivation, and implications. *Theoretical Population Biology* 118, 50–73.
- Bulmer, M. G., 1980. *The mathematical theory of quantitative genetics*. Clarendon Press, Oxford, UK.
- Cortez, M. H., Yamamichi, M., 2019. How (co)evolution alters predator responses to increased mortality: extinction thresholds and hydra effects. *Ecology* 100 (10), e02789.

- De Mazancourt, C., Johnson, E., Barraclough, T. G., 2008. Biodiversity inhibits species' evolutionary responses to changing environments. *Ecology Letters* 11 (4), 380–388.
- Deutsch, C. A., Tewksbury, J. J., Huey, B. B., Sheldon, K. S., Ghalambor, C. K., Haak, D. C., Martin, P. R., 2008. Impacts of climate warming on terrestrial ectotherms across latitude. *Proceedings of the National Academy of Sciences USA* 105, 6668–6672.
- Dieckmann, U., Doebeli, M., 1999. On the origin of species by sympatric speciation. *Nature* 400, 354–357.
- Lande, R., 1976. Natural selection and random genetic drift in phenotypic evolution. *Evolution* 30, 314–334.
- Lasky, J. R., 2019. Eco-evolutionary community turnover following environmental change. *Evolutionary Applications* 12, 1434–1448.
- Meszéna, G., 2005. Adaptive dynamics: the continuity argument. *Journal of Evolutionary Biology* 18, 1182–1185.
- Norberg, J., Urban, M. C., Vellend, M., Klausmeier, C. A., Loeuille, N., 2012. Eco-evolutionary responses of biodiversity to climate change. *Nature Climate Change* 2, 747.
URL <https://doi.org/10.1038/nclimate1588>
- Osmond, M. M., de Mazancourt, C., 2013. How competition affects evolutionary rescue. *Philosophical Transactions of the Royal Society B: Biological Sciences* 368 (1610), 20120085.
- Osmond, M. M., Otto, S. P., Klausmeier, C. A., 2017. When predators help prey adapt and persist in a changing environment. *The American Naturalist* 190 (1), 83–98.
- Otto, S. P., Day, T., 2007. *A Biologist's Guide to Mathematical Modeling in Ecology and Evolution*. Princeton University Press, Princeton, USA.
- Press, W., Teukolsky, S., Vetterling, W., Flannery, B., 2007. *Numerical Recipes 3rd Edition: The Art of Scientific Computing*. Cambridge University Press, New York, USA.
- Rackauckas, C., Ma, Y., Martensen, J., Warner, C., Zubov, K., Supekar, R., et al., 2020. Universal differential equations for scientific machine learning. arXiv.
- Robertson, A., 1968. The spectrum of genetic variation. In: Lewontin, R. C. (Ed.), *Population biology and evolution*. Syracuse University Press, Syracuse, New York, USA.
- Sadler, M., Regan, N., 2019. *Game Changer – AlphaZero's groundbreaking chess strategies and the promise of AI*. New in Chess, Alkmaar, The Netherlands.
- Schreiber, S. J., Bürger, R., Bolnick, D. I., 2011. The community effects of phenotypic and genetic variation within a predator population. *Ecology* 92, 1582–1593.
- Thomas, M. K., Kremer, C. T., Klausmeier, C. A., Litchman, E., 2012. A global pattern of thermal adaptation in marine phytoplankton. *Science* 338 (6110), 1085–1088.
- Thompson, P. L., Fronhofer, E. A., 2019. The conflict between adaptation and dispersal for maintaining biodiversity in changing environments. *Proceedings of the National Academy of Sciences USA* 116, 21061–21067.

REVIEWER COMMENTS

Reviewer #1 (Remarks to the Author):

The authors have done an admirable job of responding to reviewer comments and revisions. I find that much of the manuscript is now more clearly organized, but other parts still lack a strong message.

My main concern earlier was a lack of organization and clear narrative.

In the revision, I find the Introduction to be much improved, clear and focused.

However, I find that the Results section still lacks a clear narrative and focus. The Results could benefit from organization via sections with headings. The Results feel a bit loosely organized now.

For example, there is a paragraph discussing differences in polar, temperate, and tropical regions. I found this unexpected, because it wasn't a focus in the introduction.

I suggest parallel organizational structure from the Introduction to the other sections.

The Discussion starts with a strong organization, though it is focused on the approach, as opposed to the biological insights derived from the modeling.

As the Discussion moves into the findings from this studies, I find that it is not as clearly organized as the Introduction.

Here is a recent article modeling community-wide evolutionary response to environmental change that the authors may wish to cite

<https://www.frontiersin.org/articles/10.3389/fevo.2020.552268/full>

I mentioned Fisher's fundamental theorem with respect to the authors' finding that trait lag was lower when trait diversity was greater. The authors responded that the secondary theorem was more closely related, but I don't see it. This isn't a major point, I'm just trying to be constructive. The secondary theorem relates to trait-fitness covariance, which in these simulations isn't changing much as far as I can tell. Rather, the relationship between trait lag and trait diversity is about trait diversity – and Fisher's fundamental theorem is about the variation in fitness (admittedly not in a trait). Anyway, I don't see the point the authors make about the secondary theorem.

Minor

L17 – 'its slope and intercept' what does this refer to?

L67 – I suggest 'surprising' or 'unexpected' instead of current 'unpredictable'

L167 – 'patters'

Reviewer #2 (Remarks to the Author):

The manuscript was greatly improved through revision. However, I still have a few comments.

1. Is it possible for the authors to discuss the effects of increasing variance of temperature on community dynamics in addition to the increase of mean temperature (Fig. 2b)? Climate change may cause extreme events more often (e.g., Bathiany et al. 2018 Sci. Adv.), and it may change species coexistence (e.g., via the storage effect) and extinction dynamics (especially with the Allee effect).

2. Is it possible for the authors to upload the simulation codes to a website (e.g., github)? It will make it easier for readers to replicate the study and to modify the original model for testing their own hypotheses.

Minor comments:

L17: "slope and intercept" between what and what?

L27, 321: "surprising" for whom? It might be better to avoid the subjective expression.

L92-94: Is it possible for consumer species to have the intrinsic growth? Or is it just for resource species (at the lowest trophic level)?

L94: Is it possible for resource species to compete?

L94: "growth from consumption" does not affect resource species at the lowest trophic level, right?

L95: "loss due to being consumed" does not affect species at the highest trophic level, right?

L120-124: Are there any reasons why the authors did not consider three trophic levels?

L167: patters → patterns

L488-490, 549-551: It seems that ref. 25 and 46 are the same (Thomas et al. 2012 Science).

Reviewer #3 (Remarks to the Author):

I have read this ms for second time. Authors have made a great job in clarifying the many aspects of the ms. throughout the different sections. Below I summarize my comments from Authors' responses in the first round plus my additional comments from the new draft.

Universal differential equations

OK with the new mention in the discussion (l259-261)

Weak selection assumption

Authors explain in detail the weak selection method. Fig R3 summarizes clearly their assumptions. A few details remain however. This is a particular model with a working weak selection approximation -- but many models and fitness functions violate the weak selection assumption -- see for example <https://arxiv.org/pdf/1010.2105.pdf> -- Authors are trying to have a strong position about the non existing connection between weak selection and nearly neutral approx. I feel there is no need of such a strong position. The nearly neutral model posits a distribution of selection coefficients with a large fraction of new mutations with fitness effects near the reciprocal of population sizes. Therefore "weakly

selected” has been assumed in relation to the magnitude of $N_e \times s$ rather than to the functional or fitness effects of mutations -- so there is a clear relationship between near-neutrality, weak selection, selection coefficients and fitness beyond the continuous or discrete logistic technical details (Akashi, H., Osada, N., and Ohta, T. (2012). Weak Selection and Protein Evolution, *Genetics*, 192:15-31). For many studies (but not methods), “weakly selected” alleles could be well outside the nearly neutral range in many natural populations but then this should be explicitly stated in each study according to the nearly neutral model used to distinguish weakly selected from the nearly neutral prediction and how the fitness function connects the two. It is important to remark this explicitly in the ms. Otherwise, the reader might get confused thinking about the universality of the weak selection method without connecting it to the nearly neutral dynamics of populations.

Different selection modes

From authors' explanation, there are different selection modes that are, overall, more complicated than classic population genetics models. This goes in my previous concern -- why not drawing conclusions from a simpler, neutral landscape model to compare the outputs of it with a more complicated model -- i.e., multiple selection modes model? Notice that authors here are arguing that their model is complex -- but some of their points is about they are dealing with a "toy" model. Which is the motivation of the paper if there is no option to explore the effects of the different selection modes on biodiversity dynamics? Even if this is not possible at this time, shouldn't authors explicitly mention the different selection modes more clearly outlining they are dealing with multiple selection forces acting on the different components of their model? Please clarify specifically how the multiple selection modes enter into the fully factorial combination of fast and slow trait evolution and dispersal rates.

SIMUS

Robustness of the simus

Ok with the new number of replicates and the procedure to sample 10 out of the 100

MAJOR DRAWBACKS

0. Following the above point about the weak selection -- i would add that "the weak selection approximation is not restrictive given the model assumptions" but it is not universal so the reader should take notice of this (i.e., the weak selection approximation is assumption-dependent). Again here, I added in my previous round a note about contrasting model outputs with an "explicit" neutral model to compare the two (i.e., the neutral and the multiple selection modes model). I now see more strongly that despite the great effort made by authors in the previous round, this is more needed not only to gain robustness of the analysis, but to explore why a more complex multiple selection modes model is needed in comparison with a simpler (and faster) model.

1. Empirically supported species temperature optima. How strong it is? Examples?

Ok with the new citations

2. Climate is symmetric around the equator – how does it relate to symmetric dispersal? and what is the connection between fast-slow to large-small dispersal? Why not using a distance dispersal kernel?

Please

clarify how symmetric migration is, specifically, how m_{ij} differs from m_{ji}

Ok, I understand better the "dispersal kernel" robustness analysis introduced following Figs R4-R7 and the comparison between the rectangular and the Gaussian kernels. Is there any hope that authors discuss these two scenarios in the context of asymmetric dispersal and how symmetry can be a good proxy for the asymmetry model scenario?

3. Magic trait evolution?

Authors now emphasize that the normality assumption rules out the possibility of temperature tolerance acting as a magic trait meaning the abiotic response trait is independent of the mating or reproductive trait. However, magic traits and the mechanisms underlying them seem to be strongly supported by empirical patterns for some model systems and adaptive radiations. Authors have added a brief discussion about this point without mentioning how magic traits, which are strongly supported empirically for some model systems, alter their results? (Servedio, Maria R; Van Doorn, G Sander; Kopp, Michael; Frame, Alicia M; Nosil, Patrik (2011). Magic traits in speciation: 'magic' but not rare? *TREE*, 26:389-397).

A final remark: Authors' aim to join previous approaches mostly from pop gen in a unique framework. However, additive change is the boundary of a gradient - i.e., the many loci of small effects assumption. Gene interaction networks accounting for phenotypic change, on the other side, might range from traits governed from pure additive genetic variation to complex traits driven by epistasis and pleiotropy -- distributions markedly differ between additive and non-additive change and interaction traits are usually complex and potentially magic traits (North, T.-L. and Beaumont, M.A. (2015) Complex trait architecture: the pleiotropic model revisited. *Sci. Rep.* 5:9351; Melián CJ et al. (2018). Deciphering the interdependence between ecological and evolutionary networks. *TREE*, 33:504-512). I am not aiming to convince authors they should more convincingly show why they still keep their approach in one boundary. Please notice, authors remark their results in the top of one boundary without mentioning other more plausible scenarios from the empirical side. So, can authors go deeper with the available empirical evidence? Should authors enrich the scope of the approach by merging additive to non-additive scenarios? Mentioning briefly such another boundary could enrich the scope and broadness of the ms (Stearns, F.W. (2010) One hundred years of pleiotropy: a retrospective. *Genetics* 186:767–773; Wagner, G.P. and Zhang, J. (2011) The pleiotropic structure of the genotype–phenotype map: the evolvability of complex organisms. *Nat. Rev. Genet.* 12:204–213; North, T.-L. and Beaumont, M.A. (2015) Complex trait architecture: the pleiotropic model revisited. *Sci. Rep.* 5:9351; Pavlicev, M. and Cheverud, J. (2015) Constraints evolve: context dependency of gene effects allows evolution of pleiotropy. *Annu. Rev. Ecol. Evol. Syst.* 46:413–434; Solovieff, N. et al. (2013) Pleiotropy in complex traits: challenges and strategies. *Nat. Rev. Genet.* 14: 483–495; Melo, D. and Marroig, G. (2015) Directional selection can drive the evolution of modularity in complex traits. *Proc. Natl. Acad. Sci. U. S. A.* 112, 470–475).

Response to reviewer comments

Manuscript number: NCOMMS-20-17249A

“The importance of species interactions in eco-evolutionary community dynamics under climate change”
Anna Åkesson, Alva Curtsdotter, Anna Eklöf, Bo Ebenman, Jon Norberg & György Barabás

Reviewer 1

The authors have done an admirable job of responding to reviewer comments and revisions. I find that much of the manuscript is now more clearly organized, but other parts still lack a strong message.

My main concern earlier was a lack of organization and clear narrative. In the revision, I find the Introduction to be much improved, clear and focused. However, I find that the Results section still lacks a clear narrative and focus. The Results could benefit from organization via sections with headings. The Results feel a bit loosely organized now. For example, there is a paragraph discussing differences in polar, temperate, and tropical regions. I found this unexpected, because it wasn't a focus in the introduction. I suggest parallel organizational structure from the Introduction to the other sections.

We thank the Reviewer again for the helpful suggestions. We have now expanded and clarified the last paragraph in the Introduction, in which we state the aims of this study (lines 67-78). Exploration of the polar, temperate, and tropical regions is a part of our region-wise exploration of species' response to climate change, but we agree this was not clearly stated in the Introduction. We have now added a short section to clarify this (lines 60-64). We have also added subheadings to the Results section to improve its structure, and added a few clarifying sentences to the first paragraph of the Results (lines 170-176).

The Discussion starts with a strong organization, though it is focused on the approach, as opposed to the biological insights derived from the modeling. As the Discussion moves into the findings from this studies, I find that it is not as clearly organized as the Introduction.

We have now added subheadings to the Discussion, hopefully clarifying its structure.

Here is a recent article modeling community-wide evolutionary response to environmental change that the authors may wish to cite <https://www.frontiersin.org/articles/10.3389/fevo.2020.552268/full>

We appreciate the suggested paper and are now referencing it, both in the Introduction and Discussion.

I mentioned Fisher's fundamental theorem with respect to the authors' finding that trait lag was lower when trait diversity was greater. The authors responded that the secondary theorem was more closely related, but I don't see it. This isn't a major point, I'm just trying to be constructive. The secondary theorem relates to trait-fitness covariance, which in these simulations isn't changing much as far as I can tell. Rather, the relationship between trait lag and trait diversity is about trait diversity – and Fisher's fundamental theorem is about the variation in fitness (admittedly not in a trait). Anyway, I don't see the point the authors make about the secondary theorem.

We agree—apologies for over-complicating this. We have changed the corresponding part in our text to connect the trait lag results with Fisher's fundamental theorem, instead of the secondary theorem (lines 237-239).

L17 – 'its slope and intercept' what does this refer to?

This was indeed unclear. We have rewritten the sentence: instead of “influence its slope and intercept”, we write “influence the relationship between trait dispersion and trait lag (line 17).

L67 – I suggest ‘surprising’ or ‘unexpected’ instead of current ‘unpredictable’

We changed it to ‘unexpected’ (line 74).

L167 – ‘patters’

Fixed to ‘patterns’.

Reviewer 2

The manuscript was greatly improved through revision. However, I still have a few comments.

Once again, we thank the Reviewer for the helpful suggestions.

1. Is it possible for the authors to discuss the effects of increasing variance of temperature on community dynamics in addition to the increase of mean temperature (Fig. 2b)? Climate change may cause extreme events more often (e.g., Bathiany et al. 2018 Sci. Adv.), and it may change species coexistence (e.g., via the storage effect) and extinction dynamics (especially with the Allee effect).

As the Reviewer points out, in reality, there are annual temperature oscillations, which are expected to increase in amplitude especially in the tropical regions due to climate change (Bathiany et al. 2018). We have now added a section in the Discussion on this (lines 286-293). There we note that within-year temporal variability might alter our results, potentially promoting coexistence via temporal mechanisms like the storage effect or relative nonlinearity, but possibly leading to extinctions if Allee effects prevent species from recovering after unfavorable within-year periods. Additionally, we now also mention in our Modeling framework section that our modeled temperature increase is represented by annual averages giving a smooth increase (lines 125-126).

2. Is it possible for the authors to upload the simulation codes to a website (e.g., github)? It will make it easier for readers to replicate the study and to modify the original model for testing their own hypotheses.

Absolutely; in fact, we already had all our scripts in a public repository, though this fact might not have been advertised properly. It can be found at https://github.com/dysordys/spatial_ecoevo (see also our code availability statement in the text; lines 426-428).

Minor comments:

L17: “slope and intercept” between what and what?

This was indeed unclear. We have rewritten the sentence: “... varying species’ abilities to disperse and adapt to new temperatures influence the relationship between trait dispersion and trait lag” (lines 16-17).

L27, 321: “surprising” for whom? It might be better to avoid the subjective expression.

We have changed it to ‘unexpected’ in the first case and dropped the word altogether in the second (lines 28 and 377).

L92-94: Is it possible for consumer species to have the intrinsic growth? Or is it just for resource species (at the lowest trophic level)?

Consumers do have intrinsic growth—but it may only be negative (and it is less negative if the local temperature matches the consumer’s temperature optimum). The left panel of Figure S1 in the Supplement depicts these intrinsic mortalities. Re-reading our text here, this part in the Modeling framework section was quite confusing. So we have re-written the section (lines 97-110). This will hopefully clarify the answer to the next three points below as well.

L94: Is it possible for resource species to compete?

Yes, resource species always compete with one another. The competition coefficients can be either constant (in the baseline and trophic models) or temperature-dependent (in the temperature-dependent

competition model, with- and without a second trophic level). We now point this out explicitly (lines 104-105). On the other hand, consumer species do not interact directly, and compete only via their shared resources (line 105).

L94: “growth from consumption” does not affect resource species at the lowest trophic level, right?

Yes, that is correct (lines 109-110).

L95: “loss due to being consumed” does not affect species at the highest trophic level, right?

Correct (lines 109-110).

L120-124: Are there any reasons why the authors did not consider three trophic levels?

To start a study of the effect of species interactions on community dynamics under climate change, it pays to incorporate only as much complexity that will allow for an effective comparison between various models with- and without certain interaction types. Allowing for more general trophic network structures would be easy to implement in our model. The problem with doing so at this stage is that it opens a rich can of problems which would only distract from the main question we are after. The problems include: how should network structure be parameterized (connectance, number of species, number of trophic levels, degree of omnivory)? How to compare results across different models? What questions should one ask? (E.g., should one correlate community response with the number of top predators? Or species at intermediate trophic levels? Some other network property?) Naturally, all of the above are relevant questions and they do have answers. Those answers will in all likelihood be quite involved however. Interpreting them may be easier if there is a solid foundation to base those answers on. We hope our work provides that foundation, and leave the exploration of trophic network complexity as future work.

L167: patters to patterns

This has been fixed.

L488-490, 549-551: It seems that ref. 25 and 46 are the same (Thomas et al. 2012 Science).

Thank you for catching that, we dropped the duplicate entry.

Reviewer 3

I have read this ms for second time. Authors have made a great job in clarifying the many aspects of the ms. throughout the different sections. Below I summarize my comments from Authors' responses in the first round plus my additional comments from the new draft.

We thank the Reviewer for the thorough engagement with our work, and for the constructive comments.

Universal differential equations: OK with the new mention in the discussion (1259-261)

*Weak selection assumption: Authors explain in detail the weak selection method. Fig R3 summarizes clearly their assumptions. A few details remain however. This is a particular model with a working weak selection approximation – but many models and fitness functions violate the weak selection assumption – see for example <https://arxiv.org/pdf/1010.2105.pdf> – Authors are trying to have a strong position about the non existing connection between weak selection and nearly neutral approx. I feel there is no need of such a strong position. The nearly neutral model posits a distribution of selection coefficients with a large fraction of new mutations with fitness effects near the reciprocal of population sizes. Therefore “weakly selected” has been assumed in relation to the magnitude of $N_e \times s$ rather than to the functional or fitness effects of mutations – so there is a clear relationship between near-neutrality, weak selection, selection coefficients and fitness beyond the continuous or discrete logistic technical details (Akashi, H., Osada, N., and Otha, T. (2012). *Weak Selection and Protein Evolution, Genetics*, 192:15-31). For many studies (but not methods), “weakly selected” alleles could be well outside the nearly neutral range in many natural populations but then this should be explicitly stated in each study according to the nearly neutral model used to distinguish weakly selected from the nearly neutral prediction and how the fitness function connects the two. It is important to remark this explicitly in the ms. Otherwise, the reader might get confused thinking about the universality of the weak selection method without connecting it to the nearly neutral dynamics of populations.*

This is very true, and it is not our purpose to overstate the applicability of our approach. We have therefore added a discussion of this point (lines 294-313). We explicitly point out the connection with the nearly neutral theory, and the fact that we have to assume large effective population sizes for weak selection not to be overpowered by genetic and ecological drift. Since a new immigrant at a habitat patch will naturally have low population size, our method comes with the caveat that the approximation it uses might be undermined in such situations. We have partially controlled for this by restricting the genetic variance within very small populations (Section 3.4 in the Supplement), but that is of course more of a retrospective patch-up than a fundamental solution to the problem.

Different selection modes: From authors' explanation, there are different selection modes that are, overall, more complicated than classic population genetics models. This goes in my previous concern – why not drawing conclusions from a simpler, neutral landscape model to compare the outputs of it with a more complicated model – i.e., multiple selection modes model? Notice that authors here are arguing that their model is complex – but some of their points is about they are dealing with a “toy” model. Which is the motivation of the paper if there is no option to explore the effects of the different selection modes on biodiversity dynamics? Even if this is not possible at this time, shouldn't authors explicitly mention the different selection modes more clearly outlining they are dealing with multiple selection forces acting on the different components of their model? Please clarify specifically how the multiple selection modes enter into the fully factorial combination of fast and slow trait evolution and dispersal rates.

We have expanded our Modeling framework section, describing the different modes of selection important to our study (lines 111-118).

To address this important point better, we have re-run our simulations assuming perfect competitive equivalence of the species; please see further details in our response under “Major drawbacks 0” below. The motivation of using our more complex model instead of a simpler, neutral landscape model originates from our aim to further develop the study by Norberg et al. (2012). Their model is in a sense more “neutral”, with equal competition between species dispersing and adapting in a changing landscape. Our baseline model is similar, except we have not assumed competitive equivalence. Since our work aims to further explore the effect of species interactions in an eco-evolutionary setting, we include additional modes of selection, adding complexity such as unequal competition coefficients between resources species, a second trophic level, and temperature-dependent competition.

SIMUS

Robustness of the simus: Ok with the new number of replicates and the procedure to sample 10 out of the 100

Thank you! Please note, our results are based on all 100 replicates (except for the snapshots in Figure 3, which show a single model realization in each panel), instead of sampling 10 out of the 100.

MAJOR DRAWBACKS

0. Following the above point about the weak selection – I would add that “the weak selection approximation is not restrictive given the model assumptions” but it is not universal so the reader should take notice of this (i.e., the weak selection approximation is assumption-dependent). Again here, I added in my previous round a note about contrasting model outputs with an “explicit” neutral model to compare the two (i.e., the neutral and the multiple selection modes model). I now see more strongly that despite the great effort made by authors in the previous round, this is more needed not only to gain robustness of the analysis, but to explore why a more complex multiple selection modes model is needed in comparison with a simpler (and faster) model.

In a true neutral model, every individual would be selectively equivalent to any other, including their temperature response, and all individuals would interact equivalent independent of identity. In such a model, demographic stochasticity would be the process creating different outcomes. Since our model does not have demographic stochasticity, nothing interesting would happen if we were to transform our model into a truly neutral model. A better possibility, as the Reviewer suggests, is to compare our results with a model where all species are completely equivalent in their competitive relationships, but may still adapt differently to temperature. The inclusion of both a second trophic level and temperature-dependent competition would break this kind of competitive equivalence, so we only tested our baseline model with these equal competition coefficients.

The results are in the Supplement (Section 5.6). The same qualitative patterns are visible as in our original baseline model (Figures S28-S30). The main difference concerns local species richness, where the model with competitive equivalence gives a 25-50% reduction. In comparison with our other models, with a second trophic level and/or temperature-dependent competition, large differences in both local diversity and global richness are present.

With these new results, we achieve the disentangling of the effects of various selection modes. First, we have the model with a single trophic level and competitive equivalence—in this model there is locally no frequency-dependence, the locally best-adapted species wins, and the only process creating local diversity is migration from neighboring patches. Next, there is the same model but with competition coefficients specific to species pairs (our baseline model), and also a version of this model with no migration, which disentangles the effects of space and spatial movement from local dynamics (Supplement, Section 5.4).

We then add an extra layer of frequency-dependence, either by 1) adding a tropic level (so consumers can mediate species interactions in a frequency-dependent manner), and 2) making the competition coefficients themselves depend on temperature.

1. *Empirically supported species temperature optima. How strong it is? Examples? – Ok with the new citations*

2. *Climate is symmetric around the equator – how does it relate to symmetric dispersal? and what is the connection between fast-slow to large-small dispersal? Why not using a distance dispersal kernel? Please clarify how symmetric migration is, specifically, how m_{ij} differs from m_{ji} . – Ok, I understand better the “dispersal kernel” robustness analysis introduced following Figs R4-R7 and the comparison between the rectangular and the Gaussian kernels. Is there any hope that authors discuss these two scenarios in the context of asymmetric dispersal and how symmetry can be a good proxy for the asymmetry model scenario?*

Yes; fortunately, implementing asymmetric dispersal is straightforward. We have done so and re-ran our simulations. The asymmetry was implemented by defining the migration rate of species i from patch l to patch k as $m_i^{kl} = d_i^+$ if $k = l - 1$ (northward movement), $m_i^{kl} = d_i^-$ if $k = l + 1$ (southward movement), and 0 otherwise. We assumed first $d_i^+ > d_i^-$, (higher migration to northern locations), and second, $d_i^+ < d_i^-$ (higher migration to southern locations). We chose $d_i^+ = 1.5d_i$, $d_i^- = 0.5d_i$ in the former and $d_i^+ = 0.5d_i$, $d_i^- = 1.5d_i$ in the latter case, where d_i is the original dispersal rate of the symmetric dispersal. We included the results in the Supplement (Section 5.5). In the same section, we also included the results from using a Gaussian dispersal kernel (discussed in the previous response letter). With both the asymmetric kernels and the symmetric Gaussian kernel, the results are very similar to the original ones (Figures S19-S27). Based on these new results, it appears that kernel shape and symmetry do not affect the dynamical outcomes much.

3. *Magic trait evolution? Authors now emphasize that the normality assumption rules out the possibility of temperature tolerance acting as a magic trait meaning the abiotic response trait is independent of the mating or reproductive trait. However, magic traits and the mechanisms underlying them seem to be strongly supported by empirical patterns for some model systems and adaptive radiations. Authors have added a brief discussion about this point without mentioning how magic traits, which are strongly supported empirically for some model systems, alter their results? (Servedio, Maria R; Van Doorn, G Sander; Kopp, Michael; Frame, Alicia M; Nosil, Patrik (2011). Magic traits in speciation: ‘magic’ but not rare? TREE, 26:389-397).*

We have now added further discussion about magic traits, acknowledging that they may be common in nature (lines 314-324). We also describe possible effects of allowing for magic traits. One example is that with speciation, the currently observed decrease in species richness might be mitigated due to new species emerging.

A final remark: Authors’ aim to join previous approaches mostly from pop gen in a unique framework. However, additive change is the boundary of a gradient - i.e., the many loci of small effects assumption. Gene interaction networks accounting for phenotypic change, on the other side, might range from traits governed from pure additive genetic variation to complex traits driven by epistasis and pleiotropy – distributions markedly differ between additive and non-additive change and interaction traits are usually complex and potentially magic traits (North, T.-L. and Beaumont, M.A. (2015) Complex trait architecture: the pleiotropic model revisited. Sci. Rep. 5:9351; Melián CJ et al. (2018). Deciphering the interdependence between ecological and evolutionary networks. TREE, 33:504-512). I am not aiming to convince authors they should more convincingly show why they still keep their approach in one boundary. Please notice, authors remark their results in the top of one boundary without mentioning other more plausible scenarios from the empirical side. So, can authors go deeper with the available empirical evidence? Should authors enrich the scope of

the approach by merging additive to non-additive scenarios? Mentioning briefly such another boundary could enrich the scope and broadness of the ms (Stearns, F.W. (2010) One hundred years of pleiotropy: a retrospective. Genetics 186:767–773; Wagner, G.P. and Zhang, J. (2011) The pleiotropic structure of the genotype–phenotype map: the evolvability of complex organisms. Nat. Rev. Genet. 12:204–213; North, T.-L. and Beaumont, M.A. (2015) Complex trait architecture: the pleiotropic model revisited. Sci. Rep. 5:9351; Pavlicev, M. and Cheverud, J. (2015) Constraints evolve: context dependency of gene effects allows evolution of pleiotropy. Annu. Rev. Ecol. Evol. Syst. 46:413–434; Solovieff, N. et al. (2013) Pleiotropy in complex traits: challenges and strategies. Nat. Rev. Genet. 14: 483–495; Melo, D. and Marroig, G. (2015) Directional selection can drive the evolution of modularity in complex traits. Proc. Natl. Acad. Sci. U. S. A. 112, 470–475).

When assuming pure additive genetic variation using a quantitative genetic approximation, the total phenotypic variance is simply the sum of the additive variance and the environmental variance: $V_P = V_A + V_E$. In the general case, phenotypic variance also includes dominance variance and epistatic variance: $V_P = V_A + V_D + V_I + V_E$. Our additive assumption is indeed the boundary of a continuum, where we assume that the interaction network of genes is especially simple—that is, everything is just additive and there are no interaction effects between genes. This is indeed a strong assumption about the genetics of the organisms. Nevertheless, this has proven to be a good starting point for understanding selection phenomena in a variety of systems, such as the evolution of abdominal bristle count in *Drosophila melanogaster* (Clayton et al. 1957), or oil and protein content in corn (*Zea mays*; Dudley 2007). One can therefore reasonably hope that our bare-bones genetic model is not too far off the mark for describing real evolutionary processes.

That said, in reality there are interaction effects such as pleiotropy, epistasis, and dominance. When considering a whole gene regulatory network in all its genuine complexity, these effects can be very important, and can change the expected outcomes from those we have observed. We now acknowledge this in the Discussion (lines 294-299).

References

- Bathiany, S., Dakos, V., Scheffer, M., Lenton, T. M., 2018. Climate models predict increasing temperature variability in poor countries. *Science Advances* 4 (5).
- Clayton, G. A., Morris, J. A., Robertson, A., Feb. 1957. An experimental check on quantitative genetical theory I. Short-term responses to selection. *Journal of Genetics* 55 (1), 131–151.
- Dudley, J. W., 2007. From means to QTL: the Illinois Long-Term Selection Experiment as a case study in quantitative genetics. *Crop Science* 47 (S3), S20–S31.
- Norberg, J., Urban, M. C., Vellend, M., Klausmeier, C. A., Loeuille, N., 2012. Eco-evolutionary responses of biodiversity to climate change. *Nature Climate Change* 2, 747.
URL <https://doi.org/10.1038/nclimate1588>

REVIEWER COMMENTS

Reviewer #1 (Remarks to the Author):

The authors have done a nice job of responding to previous comments. The manuscript is a strong contribution.

Reviewer #2 (Remarks to the Author):

I was impressed by the authors' effort for improving the manuscript, but I still have a few comments.

Temperature-dependent competition and trophic interactions (1) reduces the species loss (Figure 4) and (2) reduces the trait dispersion and increases the community average trait lag (Figure 7). How are the two results related? Did the authors check a negative correlation between the number of species and trait dispersion? What is the mechanism behind the pattern? Why is the pattern (i.e., higher species richness and larger trait lag) different from other BEF (Biodiversity and Ecosystem Functioning) studies?

L12-14: Does the negative correlation between the trait dispersion and trait lag exist in the previous studies (Norberg et al. 2012, Lasky 2019)? If so, why did not the previous studies examine the relationship?

L91: "heritability reduction" sounds vague and confusing as heritability is $V_g/(V_g+V_e)$ and it is possible to reduce it by increasing V_e instead of reducing V_g .

L104-105: Is competition between resource species interference competition? Or exploitative competition as competition between consumers?

L153-154: Did the authors consider beta diversity (= alpha/gamma) explicitly in this study?

L200-202: Is it useful to analyze the results by 2-way ANOVA for examining the interaction of temperature-dependent competition and consumption?

L304: effective community size → effective population size?

L319: Having magic traits is not a necessary condition for speciation. Even "pseudomagic traits" can promote speciation (Servedio & Bürger 2020 Evolution). Also, non-ecological (mutation-order) speciation may promote divergence in the timescale considered in this study.

Response to reviewer comments

Manuscript number: NCOMMS-20-17249C

“The importance of species interactions in eco-evolutionary community dynamics under climate change”
Anna Åkesson, Alva Curtsdotter, Anna Eklöf, Bo Ebenman, Jon Norberg & György Barabás

Reviewer 1

The authors have done a nice job of responding to previous comments. The manuscript is a strong contribution.

We thank the Reviewer again for the encouragement, and for all the work dedicated to improving our manuscript.

Reviewer 2

I was impressed by the authors' effort for improving the manuscript, but I still have a few comments.

We are very grateful for the Reviewer's continued engagement with our work—and especially so for the main comment we have received below. Getting to the bottom of that issue has revealed an important miscommunication between the authors, which has led to us adopting a faulty definition for the community trait lag (see below). Fixing this issue has meant that, while the overall negative relationship between trait lag and trait dispersion is retained, the results are now much more streamlined. We apologize for the mistake, and thank the Reviewer again for correctly intuiting that something was amiss; this has saved us from disseminating flawed results!

Temperature-dependent competition and trophic interactions (1) reduces the species loss (Figure 4) and (2) reduces the trait dispersion and increases the community average trait lag (Figure 7). How are the two results related?

Trying to dig to the bottom of these questions made us discover that we were using the wrong formula for the trait lag. The original (incorrect) definition was $\mathcal{A}^k = \sum_{i=1}^S n_i^k (T^k - \mu_i^k)^2$, the weighted sum of squared deviations of species' local trait values μ_i^k from the local temperature T^k . Instead, as the trait driver theory expert in our group (Jon Norberg) corrected us, the square is not needed: $\mathcal{A}^k = \sum_{i=1}^S n_i^k (T^k - \mu_i^k)$. Since the n_i^k are the relative densities in patch k which sum to 1, $\sum_{i=1}^S n_i^k T^k = T^k$. Then, using the notation $\bar{\mu}^k = \sum_{i=1}^S n_i^k \mu_i^k$, one can simply write $\mathcal{A}^k = T^k - \bar{\mu}^k$. We have now fixed this in the Methods (Eq. 11) and the Supplement (Eq. S67).

The trait lag versus dispersion results are now re-generated with the correct formula for trait lag. While the overall negative relationship between the two is retained, the data points form transparently distinct clusters based on model type and parameterization. Temperature-dependent competition clearly reduces trait lag and increases trait dispersion. Additionally, with temperature-dependent competition, fast dispersal further increases dispersion and decreases trait lag. The opposite holds for slow dispersal.

To account for these changes, we have updated Figures 7 and S41 (the original trait lag figure in the main text plus the corresponding figure for 30 species in the Supplement), and adjusted the “Global trends” section of the Results (lines 233-253), the “Trait lag and trait dispersion” subheading of the Discussion (lines 383-402), and the Abstract (lines 12-16).

Did the authors check a negative correlation between the number of species and trait dispersion?

As the new results are much more aligned with common sense (temperature-dependent competition reduces, instead of increases, trait lag), we no longer expect a negative correlation. That said, it is a very good idea to check this relationship anyway, which we have done in the new Section 4.1 of the Supplement. In brief, there is an overall positive relationship between richness and trait dispersion—but this is sometimes reversed for disaggregated pieces of the data, e.g. for temperature-dependent competition with low genetic variance and dispersal ability (Figure S4).

What is the mechanism behind the pattern?

In outline, this again has become much more straightforward to interpret: under temperature-dependent competition, which effectively allows species to niche-differentiate along temperature as a trait axis, species can space out approximately evenly around local temperatures T^k , making the mean traits $\bar{\mu}^k$ nearly equal to the local temperatures and thus creating a low trait lag $\mathcal{A}^k = T^k - \bar{\mu}^k$. We discuss this in lines 233-253 in the Results and 383-402 in the Discussion.

Why is the pattern (i.e., higher species richness and larger trait lag) different from other BEF (Biodiversity and Ecosystem Functioning) studies?

While the negative relationship between dispersion and trait lag is maintained, we now also have an overall negative correlation between species richness and trait lag as well (Figures 7 and S4). This is certainly consistent with much of the BEF literature: more trait diversity allows better response to changing environments. The difference is that most BEF studies focus on total biomass or resource consumption as the main ecosystem function of interest, whereas we look at the community's ability to respond to climate change as a function of its trait diversity. (See also lines 403-411 in the Discussion.)

L12-14: Does the negative correlation between the trait dispersion and trait lag exist in the previous studies (Norberg et al. 2012, Lasky 2019)? If so, why did not the previous studies examine the relationship?

Jon Norberg, one of us who was also the lead author of Norberg et al. (2012), says that this did not come up at the time. Whether Jesse Lasky (2019) was aware of this expected relationship when he wrote his paper, probably only he knows. In either case, the relationship was not investigated, and so while it is likely that it would be observed, we cannot say for certain.

L91: "heritability reduction" sounds vague and confusing as heritability is $V_g/(V_g + V_e)$ and it is possible to reduce it by increasing V_e instead of reducing V_g .

Good point—we have changed the text to “reduction in genetic variance” (lines 89-90).

L104-105: Is competition between resource species interference competition? Or exploitative competition as competition between consumers?

Competition between resource species is phenomenological: we simply prescribed competition coefficients directly between the resource species, without specifying the mechanism behind it. However, since the competitive interactions have Lotka–Volterra form, one can always imagine that they arise from competition for a set of noninteracting resources whose dynamics are fast compared with the consumers' dynamics (where, in this context, it is our *resource* species which play the role of consumers.) See e.g. MacArthur (1970) or Pastore et al. (2021, Eqs. S7-S11) for deriving the Lotka–Volterra from such a consumer-resource system. If this is the favored interpretation for how the competition terms arise, then competition is strictly exploitative. We now specify this in our text (lines 102-104).

L153-154: Did the authors consider beta diversity (= alpha/gamma) explicitly in this study?

Yes, this was already requested in the first round of the review process, so we have added the results to the Supplement (Section 5.2).

L200-202: Is it useful to analyze the results by 2-way ANOVA for examining the interaction of temperature-dependent competition and consumption?

This is a very good idea—as long as we use the results only to estimate effect sizes and avoid inferring significance (since the data are computer-generated, low P -values can be guaranteed; White et al. 2014). We have now included the results from the 2-way ANOVA (Supplement, Section 5.4), applied to local diversity at the final $t = 2500$. This was done separately for each each combination of parameter settings (high and low genetic variance and dispersal ability). To obtain estimated effect size differences between various treatments (presence or not of either trophic interactions or temperature-dependent competition), we have further fed the ANOVA results to a Tukey test. This indeed reveals that the individual effect of trophic structure and temperature-dependent competition are both strong when compared to the baseline model. However, their joint action does not create much of an effect beyond what temperature-dependence can individually produce.

L304: effective community size \rightarrow effective population size?

Fixed.

L319: Having magic traits is not a necessary condition for speciation. Even "pseudomagic traits" can promote speciation (Servedio and Bürger 2020 Evolution). Also, non-ecological (mutation-order) speciation may promote divergence in the timescale considered in this study.

We thank the Reviewer for pointing out the recent contribution by Servedio and Bürger (2020). We now mention in our text that speciation may also be promoted by pseudomagic traits (e.g., two tightly linked loci, one under divergent selection and the other acting as a mating cue), as well as non-ecological modes of speciation (lines 319-325).

References

- Lasky, J. R., 2019. Eco-evolutionary community turnover following environmental change. *Evolutionary Applications* 12, 1434–1448.
- MacArthur, R. H., 1970. Species packing and competitive equilibria for many species. *Theoretical Population Biology* 1, 1–11.
- Norberg, J., Urban, M. C., Vellend, M., Klausmeier, C. A., Loeuille, N., 2012. Eco-evolutionary responses of biodiversity to climate change. *Nature Climate Change* 2, 747.
URL <https://doi.org/10.1038/nclimate1588>
- Pastore, A. I., Barabás, G., Bimler, M. D., Mayfield, M. M., Miller, T. E., 2021. The evolution of niche overlap and competitive differences. *Nature Ecology & Evolution* 5, 330–337.
- Servedio, M. R., Bürger, R., 2020. The effectiveness of pseudomagic traits in promoting divergence and enhancing local adaptation. *Evolution* 74 (11), 2438–2450.
- White, J. W., Rassweiler, A., Samhouri, J. F., Stier, A. C., White, C., 2014. Ecologists should not use statistical significance tests to interpret simulation model results. *Oikos* 123, 385–388.

REVIEWERS' COMMENTS

Reviewer #2 (Remarks to the Author):

The authors have done a great job of revising the manuscript. I have a few additional comments.

L68: co-existence → coexistence

L103: I am not sure what "direct competition" means. "Interference competition" may be a better antonym of "exploitative competition" according to the Princeton Guide to Ecology (Glossary of II.6: Competition and Coexistence in Animal Communities).

L231: It would be great if the authors can add a citation of extinction debt (maybe Tilman et al. 1994 Nature?).

L302: effective community size → effective population size

L323-324: It would be great if the authors can add a citation of non-ecological (mutation-order) speciation.

L340: Is "reduced interspecific competition with increasing trait distance" equivalent to ecological character displacement?

L511: rescuetheory → rescue theory

Reviewer #3 (Remarks to the Author):

I have read this ms again and again and it has been improving more and more at each stage.

I congratulate authors for communicating all the new simus-updates to the reviewers in such a detailed and deep way and also in how they have been discussing among them to improve the flow and the robustness of the approach. I certainly appreciate deep clarifications around their weak selection method, the extended species equivalence scenario to distinguish competitive equivalence, non-equivalence, and frequency-dependent selection, the new simus with asymmetric dispersal, the discussion of the additive genetic variation using a quantitative genetic approximation as a boundary continuum, the corrections for the trait-lag formulas, and so on. The ms. has gain in robustness along the way and I am happy that authors have had the patience to make it happen.

Response to reviewer comments

Manuscript number: NCOMMS-20-17249C

“The importance of species interactions in eco-evolutionary community dynamics under climate change”
Anna Åkesson, Alva Curtsdotter, Anna Eklöf, Bo Ebenman, Jon Norberg & György Barabás

Reviewer 2

The authors have done a great job of revising the manuscript. I have a few additional comments.

We thank the Reviewer again for the encouragement, and for all the work dedicated to improving our manuscript.

L68: co-existence → coexistence

Changed.

L103: I am not sure what "direct competition" means. "Interference competition" may be a better antonym of "exploitative competition" according to the Princeton Guide to Ecology (Glossary of II.6: Competition and Coexistence in Animal Communities).

We agree, and have rewritten the phrase.

L231: It would be great if the authors can add a citation of extinction debt (maybe Tilman et al. 1994 Nature?).

That is a good idea—we have added the reference.

L302: effective community size → effective population size

Changed.

L323-324: It would be great if the authors can add a citation of non-ecological (mutation-order) speciation.

We have added a reference to Schluter (2009).

L340: Is "reduced interspecific competition with increasing trait distance" equivalent to ecological character displacement?

There is a subtle difference between the two. Ecological character displacement is the process whereby two species diverge in their traits in response to competition. This process is often indeed driven by decreasing competition with increasing trait distance. But the two are not equivalent, and decreasing competition with increasing trait distance does not imply that ecological character displacement will occur. For example, if there is no standing genetic variation or trait plasticity in the species, then there can be no evolution, and therefore character displacement is not an option. For this reason, it is more accurate here to use the phrase “reduced interspecific competition with increasing trait distance” than “ecological character displacement”.

L511: rescuetheory → rescue theory

We have fixed the typo.

Reviewer 3

I have read this ms again and again and it has been improving more and more at each stage.

I congratulate authors for communicating all the new simus-updates to the reviewers in such a detailed and deep way and also in how they have been discussing among them to improve the flow and the robustness of the approach. I certainly appreciate deep clarifications around their weak selection method, the extended species equivalence scenario to distinguish competitive equivalence, non-equivalence, and frequency-dependent selection, the new simus with asymmetric dispersal, the discussion of the additive genetic variation using a quantitative genetic approximation as a boundary continuum, the corrections for the trait-lag formulas, and so on. The ms. has gain in robustness along the way and I am happy that authors have had the patience to make it happen.

We thank the Reviewer for the encouraging words, and are pleased to hear that the changes made following Reviewer comments are to his/her satisfaction.

References

Schluter, D., 2009. Evidence for ecological speciation and its alternative. *Science* 323 (5915), 737–741.